# Cellular mechanisms underlying central sensitization in a mouse model of chronic muscle pain

Yu-Ling Lin[1], Zhu-Sen Yang[1], Wai-Yi Wong[1], Shih-Che Lin[1], Shuu-Jiun Wang[1,2,3,4], Shih-Pin Chen[1,2,3,4,5,6], Jen-Kun Cheng[7,8], Hui Lu[9], Cheng-Chang Lien[1,2]*

[1]Institute of Neuroscience, National Yang Ming Chiao Tung University, Taipei, Taiwan; [2]Brain Research Center, National Yang Ming Chiao Tung University, Taipei, Taiwan; [3]Department of Neurology, Neurological Institute, Taipei Veterans General Hospital, Taipei, Taiwan; [4]Faculty of Medicine, National Yang Ming Chiao Tung University, Taipei, Taiwan; [5]Institute of Clinical Medicine, National Yang Ming Chiao Tung University, Taipei, Taiwan; [6]Division of Translational Research, Department of Medical Research, Taipei Veterans General Hospital, Taipei, Taiwan; [7]Department of Medicine, MacKay Medical College, New Taipei, Taiwan; [8]Department of Anesthesiology, MacKay Memorial Hospital, Taipei, Taiwan; [9]Department of Pharmacology and Physiology, George Washington University, Washington, United States

**Abstract** Chronic pain disorders are often associated with negative emotions, including anxiety and depression. The central nucleus of the amygdala (CeA) has emerged as an integrative hub for nociceptive and affective components during central pain development. Prior adverse injuries are precipitating factors thought to transform nociceptors into a primed state for chronic pain. However, the cellular basis underlying the primed state and the subsequent development of chronic pain remains unknown. Here, we investigated the cellular and synaptic alterations of the CeA in a mouse model of chronic muscle pain. In these mice, local infusion of pregabalin, a clinically approved drug for fibromyalgia and other chronic pain disorders, into the CeA or chemogenetic inactivation of the somatostatin-expressing CeA (CeA-SST) neurons during the priming phase prevented the chronification of pain. Further, electrophysiological recording revealed that the CeA-SST neurons had increased excitatory synaptic drive and enhanced neuronal excitability in the chronic pain states. Finally, either chemogenetic inactivation of the CeA-SST neurons or pharmacological suppression of the nociceptive afferents from the brainstem to the CeA-SST neurons alleviated chronic pain and anxio-depressive symptoms. These data raise the possibility of targeting treatments to CeA-SST neurons to prevent central pain sensitization.

*For correspondence:
cclien@ym.edu.tw

## Editor's evaluation

This study investigates the role of the central amygdala (CeA) on the mechanisms underlying chronic pain. An acid-induced muscle pain (AIMP) mouse model is used. Changes of excitatory synaptic and intrinsic excitability of GABAergic somatostatin-expressing (SST+) neurons of the CeA are detected in the chronic MP model. The manuscript is extensive, well-illustrated and contains valuable findings that are likely to inspire future work to progress the understanding of the GABAergic system on patho-physiological events.

## Introduction

Prior adverse events or perceived physical injuries are one of the precipitating factors for chronic pain (*Mills et al., 2019*; *Stevans et al., 2021*). A prior injury is thought to transform nociceptors into a primed state, which lasts variably in different types of chronic pain disorders (*Kandasamy and Price, 2015*; *Reichling and Levine, 2009*; *Sun and Chen, 2016*). Subsequent injuries occurring during the priming phase can result in the chronification of pain (*Sun and Chen, 2016*). Acid-induced muscle pain (MP) in rodents, which is a preclinical model for chronic MP disorders, including fibromyalgia syndrome, requires two episodes of acute MP induction (*Sluka et al., 2001*). Specifically, the second acid injection into the gastrocnemius muscle of animals during a priming phase is warranted for the development of chronic MP (*Chen et al., 2014*; *Sluka and Clauw, 2016*; *Sun and Chen, 2016*). Chronic pain disorders are associated with psychiatric comorbidities (*Hooten, 2016*; *Velly and Mohit, 2018*). Indeed, patients with fibromyalgia syndrome have chronic widespread pain and other comorbid symptoms, for example, sleep, psychological, and cognitive disturbances (*Clauw, 2014*; *Sluka and Clauw, 2016*).

Central sensitization is believed to underlie widespread MP and affective symptoms in the fibromyalgia syndrome (*Clauw, 2014*; *Woolf, 2011*). However, the cellular basis underpinning central sensitization in MP disorders is poorly elucidated. Among the anatomically and functionally distinct amygdaloid nuclei (*Duvarci and Pare, 2014*; *LeDoux, 2000*), the central nucleus of the amygdala (CeA) has emerged as an integrative hub for nociceptive and affective components during the development of chronic pain states (*Kuner and Kuner, 2021*; *Neugebauer et al., 2004*; *Thompson and Neugebauer, 2017*). The CeA receives a direct nociceptive projection from the parabrachial nucleus (PBN). Maladaptive changes in synaptic transmission at the PBN to the CeA neuron synapses are thought to underlie persistent pathological pain states (*Ikeda et al., 2007*; *Li and Sheets, 2020*; *Wilson et al., 2019*). The CeA houses distinct gamma aminobutyric acid (GABA)-ergic neuronal populations; the two major groups include the somatostatin-expressing (SST) and protein kinase C delta-expressing (PKCδ) neurons (*Babaev et al., 2018*; *Haubensak et al., 2010*; *Hou et al., 2016*). These two types of neurons form a reciprocal inhibitory circuit (*Babaev et al., 2018*; *Ciocchi et al., 2010*; *Haubensak et al., 2010*). It is believed that PKCδ neurons are pro-nociceptive, whereas SST neurons are antinociceptive under physiological conditions (*Wilson et al., 2019*). However, the role of these two types of neurons in the regulation of nociception is inconsistent in different chronic pain models (*Chen et al., 2022*; *Wilson et al., 2019*; *Zhou et al., 2019*).

Here, we aimed to investigate the cellular basis underlying MP using the acid-induced MP model. By combining electrophysiology, chemogenetics, and in vivo calcium imaging, we attempted to demonstrate that CeA-SST neurons, which are believed to be antinociceptive neurons, become hyperexcitable following MP development. Consistent with this finding, selective chemogenetic inactivation of CeA-SST neurons or suppression of synaptic transmission of the PBN to the CeA-SST neurons alleviated MP and negative emotions.

## Results

### Intra-CeA application of pregabalin alleviated pain in an MP mouse model

Acid-induced MP in rats or mice is considered a preclinical model of fibromyalgia (*Cheng et al., 2011*; *Min et al., 2011*; *Sluka et al., 2001*). In this study, we induced MP in mice using a well-established protocol for acid-induced MP. Mice with acidic (pH 4.0) saline injected into the gastrocnemius muscle unilaterally on day 0 (baseline [BL]) and day 3 (*Figure 1A*) are referred to as the MP mice. Following the first acidic saline injection, the MP mice showed a transient decrease in the paw withdrawal (PW) threshold in both the ipsilateral ('ipsi') and contralateral ('contra') hind limbs in response to the von Frey filament stimulation (*Figure 1B*). The acute pain mostly recovered by day 3 (*Figure 1B*). A second injection, however, caused a sustained decrease in the PW threshold that lasted for at least 14 days (*Figure 1B*). In contrast to the MP mice, the neutral (pH 7.2) saline-injected mice, referred to as the control (Ctrl) mice, showed no significant changes in the PW response to the von Frey filament stimulation (*Figure 1B*). Collectively, only the MP mice exhibited an increase in the PW response to a wide range of von Frey filament stimulations bilaterally (*Figure 1C*). The mechanical hypersensitivity induced in the MP mice lasted for at least 2 weeks (*Figure 1B and C*).

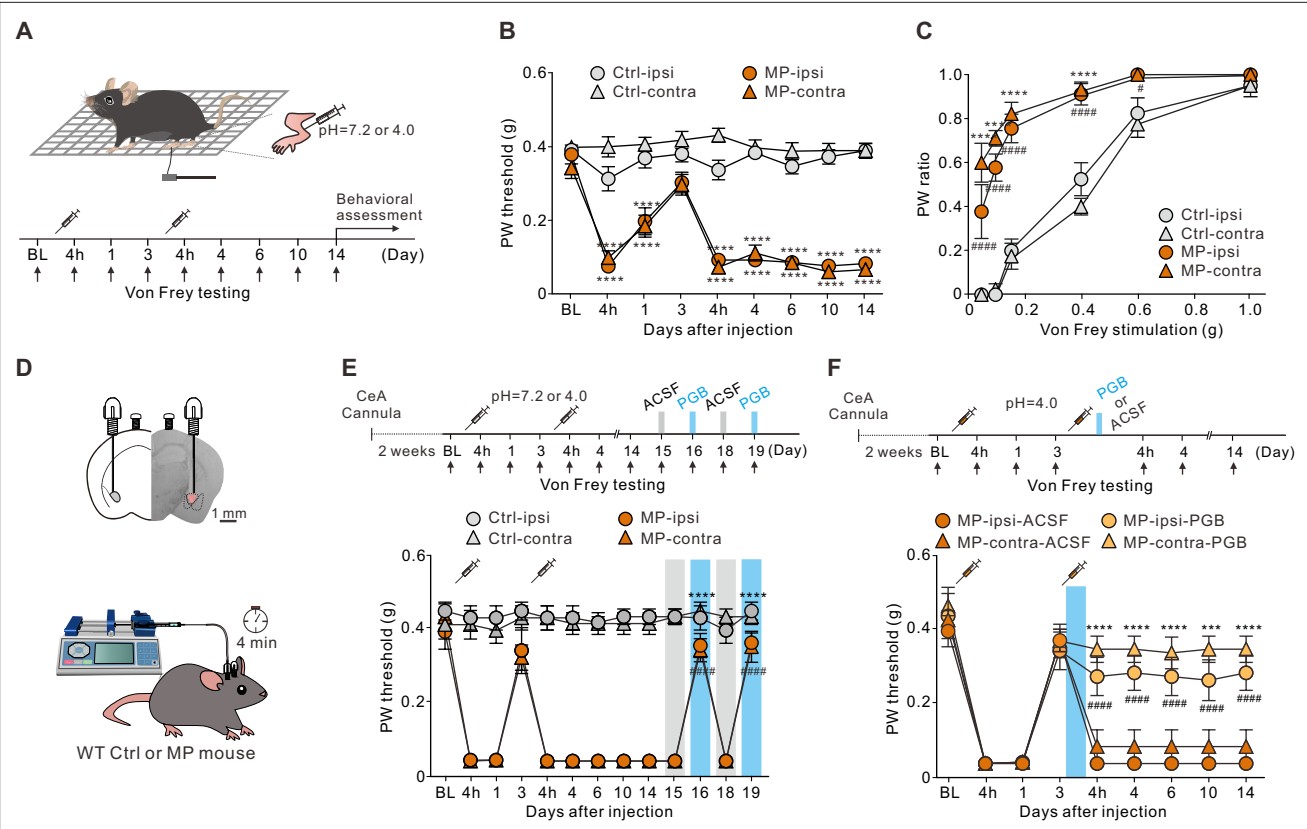

**Figure 1.** Local application of pregabalin (PGB) in the central nucleus of the amygdala (CeA) alleviated pain in a muscle pain (MP) mouse model. (**A**) Schematic of MP induction protocol and experimental timeline. Mice were injected with either neutral (pH 7.2, Ctrl mice) or acidic (pH 4.0, MP mice) saline into the gastrocnemius muscle unilaterally on days 0 and 3. (**B**) Plot of the paw withdrawal (PW) threshold to the mechanical stimuli over time (Ctrl, n=26; MP, n=28; two-way analysis of variance (ANOVA) with Tukey's post hoc test, F(8,936) = 30.97, ****p<0.0001 relative to baseline [BL]). (**C**) The bilateral PW responses to different filaments on day 14 in the Ctrl and MP mice (Ctrl, n=8; MP, n=9; two-way ANOVA with Tukey's post hoc test, F(3,180) = 121.1, #p<0.05, ****,####p<0.0001; * indicates the comparison between ipsilateral hindpaws; # indicates the comparison between contralateral hindpaws). (**D**) Representative image of the PGB infusion site (labeled with a red fluorescent dye, sulforhodamine 101) and experimental schematic. (**E**) Effects of PGB treatment on the PW threshold in the Ctrl and MP mice (Ctrl, n=13; MP, n=12; two-way ANOVA with Tukey's post hoc test, F(12,598) = 38.2, ****,####p<0.0001 relative to day 15; * indicates the comparison between the ipsilateral hindpaws; # indicates the comparison between the contralateral hindpaws. The blue area indicates the period of PGB treatment). (**F**) Top, experimental timeline. Right after the second injection of the acidic saline, one of the mouse groups was infused with PGB, and the other with artificial cerebrospinal fluid (ACSF). Bottom, effects of PGB treatment on the PW threshold in the ACSF and PGB groups (ACSF, n=8; PGB, n=9; two-way ANOVA with Tukey's post hoc test, F(3,270) = 48.7, ***p<0.001, ****,####p<0.0001; * indicates the comparison between the ipsilateral hindpaws; # indicates the comparison between the contralateral hindpaws. The blue area indicates the period of PGB treatment).

The online version of this article includes the following source data and figure supplement(s) for figure 1:

**Source data 1.** Numerical data to support the graphs in *Figure 1*.

**Figure supplement 1.** Comorbid affective symptoms in a muscle pain (MP) mouse model.

**Figure supplement 1—source data 1.** Numerical data to support the graphs in *Figure 1—figure supplement 1*.

**Figure supplement 2.** Expression of α2δ in the parabrachial nucleus (PBN).

In addition to mechanical allodynia, the mice with MP showed various affective symptoms (*Figure 1—figure supplement 1*). The elevated plus maze (EPM) and light/dark (L/D) box are common approach-avoidance conflict tests based on the general aversion of mice to bright and open environments (*Calhoon and Tye, 2015*; *Walf and Frye, 2007*). Anxious mice prefer to stay in the closed arms and dark compartment. Compared to the Ctrl mice, the MP mice spent less time in the open arms (*Figure 1—figure supplement 1A*) and made fewer transitions between the light and dark zones in the L/D box test (*Figure 1—figure supplement 1B*). Neither of these behavioral phenotypes was due to reduced locomotion because the total travel distances of the MP and Ctrl mice were

not significantly different (*Figure 1—figure supplement 1A, B*). We also tested mice in the marble burying test where anxious animals directed their energies toward minimizing threatening stimuli and tended to bury more marbles. Compared to the Ctrl mice, the MP mice buried more marbles in the marble burying test (*Figure 1—figure supplement 1C*).

Anxiety disorders are often associated with depression-like behavior and social avoidance (*Allsop et al., 2014*). The forced swim test (FST) is commonly used for testing depression-like behavior in mice. Compared to the Ctrl mice, the MP mice spent more time in the floating state, suggesting enhanced depression-like behavior (*Figure 1—figure supplement 1D*). Finally, we tested the sociability of the mice by measuring the time spent in a social chamber containing a novel mouse (*Yang et al., 2011*). Accordingly, the MP mice spent less time in the social chamber than the Ctrl mice (*Figure 1—figure supplement 1E*). The total travel distance of the MP mice was not significantly different from that of the Ctrl mice (*Figure 1—figure supplement 1E*). Taken together, these results suggest that the MP mice exhibited mechanical allodynia and hyperalgesia as well as a variety of behavioral traits commonly associated with anxiety and depression. Thus, the acid-induced MP model satisfies the face validity of chronic MP (*Calhoon and Tye, 2015*).

Therefore, we tested the predictive validity of this model. Pregabalin (PGB) is a Food and Drug Administration-approved drug for the treatment of chronic pain disorders and affective comorbidities. PGB is a selective ligand for the α2δ subunit of voltage-gated calcium channels (VGCCs; *Häuser et al., 2009*). However, the exact action site of PGB on both the sensory and affective dimensions is unknown. The CeA receives a direct nociceptive projection from the PBN, which is enriched with the calcium channel α2δ subunit (*Cole et al., 2005*; *Figure 1—figure supplement 2*). To test whether PGB acted on the PBN-CeA pathway, we locally infused PGB into the CeA of the Ctrl and MP mice bilaterally through a cannula (*Figure 1D*). Compared to the Ctrl mice, cannula infusion of PGB (1 mM, 0.15 µL/site), instead of a vehicle (artificial cerebrospinal fluid [ACSF]), into the CeA of the MP mice on days 16 and 19 increased the PW threshold (*Figure 1E*). This result supports the predictive validity of this model.

In this model, the first acidic saline injection initiated a priming phase, which reportedly lasts for 5–8 days (*Chen et al., 2014*; *Sun and Chen, 2016*). Therefore, we determined whether earlier administration of PGB could prevent chronic pain development. Thus, we infused ACSF or PGB bilaterally through a cannula on day 3 into the CeA of the MP mice right after the second acidic saline injection (*Figure 1F*). The group infused with ACSF was able to develop chronic pain effectively. In contrast, the pain response of the group infused with PGB became weakened (*Figure 1F*). This result demonstrated that local infusion of PGB into the CeA during hyperalgesic priming was sufficient to prevent the development of chronic pain.

## Local infusion of PGB into the CeA alleviated negative emotions

To test whether local PGB administration in the CeA also alleviated negative emotions in the MP mice, we infused PGB into the CeA of the Ctrl and MP mice bilaterally through a cannula (*Figure 2A*). Cannula infusion of PGB (1 mM, 0.15 µL/site), instead of a vehicle (ACSF), into the CeA increased the open-arm time (*Figure 2B*) in the EPM test and decreased the number of buried marbles in the marble burying test (*Figure 2C*) in the MP mice compared to the Ctrl mice. In addition, we tested depression-like behavior and sociability of the mice using the FST and three-chambered social test. Local infusion of PGB into the CeA decreased the immobility time in the FST (*Figure 2D*) and increased the social-zone time in the three-chambered social test (*Figure 2E*) in the MP mice compared the Ctrl mice. Notably, PGB at the dosage used in this study had no effect on locomotion (*Figure 2B and E*) and had little effect on the affective behaviors in the Ctrl mice (*Figure 2B–E*). These results indicated that local infusion of PGB in the CeA in chronic pain conditions alleviated negative emotions.

Considering that the PGB treatment during the priming phase is sufficient for preventing chronic pain development, we also tested whether PGB delivery during the priming phase can prevent comorbid affective symptoms. Thus, we infused ACSF or PGB into the CeA of the MP mice bilaterally through a cannula right after the second acidic saline injection (*Figure 2F*). The same behavior assessments were performed in this experiment on day 14 (*Figure 2F*). Cannula infusion of PGB in the primed state increased the exploration time in the open arms (*Figure 2G*), decreased the number of marbles buried (*Figure 2H*), decreased the immobility time (*Figure 2I*), and increased the social-zone time (*Figure 2J*) in the MP mice compared to the ACSF group. Taken together, the infusion of PGB

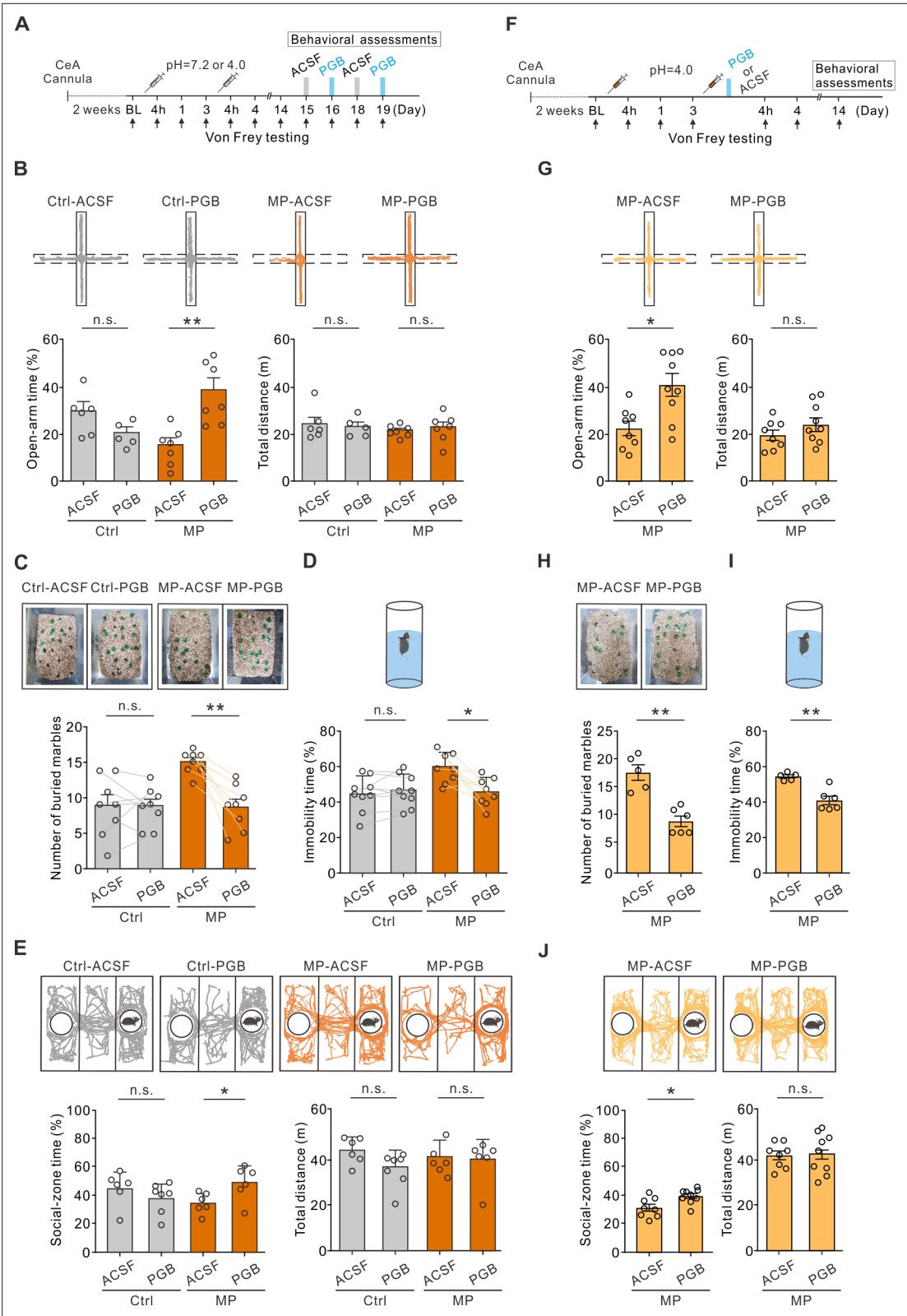

**Figure 2.** Local infusion of pregabalin (PGB) in the central nucleus of the amygdala (CeA) alleviated negative emotions. (**A**) Experimental timeline. (**B**) Top, representative travel trajectories of each Ctrl and muscle pain (MP) group during the elevated plus maze (EPM) test. Bottom, summary of the effects of PGB treatment on open-arm time (Ctrl: artificial cerebrospinal fluid [ACSF], 29.6 ± 3.8%, n=6; PGB, 20.0 ± 2.7%, n=5; Mann–Whitney test, U=4, n.s., non-significant, p=0.052. MP: ACSF, 15.1 ± 3.0%, n=7; PGB, 38.4 ± 5.1%, n=7; Mann–Whitney test, U=2, **p=0.002) and total distance (Ctrl: ACSF,

*Figure 2 continued on next page*

*Figure 2 continued*

24.1±2.9 m, n=6; PGB, 22.9±2.1 m, n=5; Mann–Whitney test, U=14, n.s., non-significant, p=0.875. MP: ACSF, 21.4±1.0 m, n=7; PGB, 22.7±2.3 m, n=7; Mann–Whitney test, U=18, n.s., non-significant, p=0.446). (**C**) Summary of the effects of PGB treatment on the number of buried marbles (Ctrl: ACSF, 8.9±1.5, n=8; PGB, 8.8±1.0, n=8; Wilcoxon matched-pairs signed rank test, n.s., non-significant, p=0.938. MP: ACSF, 15.0±0.6, n=8; PGB, 8.6±1.1, n=8; Wilcoxon matched-pairs signed rank test, **p=0.008). (**D**) Summary of relative time of immobility in the forced swim test (FST) (Ctrl: ACSF, 44.2 ± 3.3%, n=9; PGB, 46.1 ± 3.0%, n=9; Wilcoxon matched-pairs signed rank test, n.s., non-significant, p=0.734. MP: ACSF, 57.4 ± 2.4%, n=8; PGB, 46.0 ± 2.8%, n=8; Wilcoxon matched-pairs signed rank test, *p=0.023). (**E**) Top, representative travel trajectories of each Ctrl and MP group during the three-chamber sociability test. Bottom, summary of the effects of PGB treatment on social-zone time (Ctrl: ACSF, 44.6 ± 4.8%, n=6; PGB, 37.8 ± 3.8%, n=7; Mann–Whitney test, U=10, n.s., non-significant, p=0.138. MP: ACSF, 34.2 ± 2.9%, n=6; PGB, 48.6 ± 5.0%, n=6; Mann–Whitney test, U=5, *p=0.041) and total distance (Ctrl: ACSF, 42.3±2.3 m, n=6; PGB, 35.2±2.7 m, n=7; Mann–Whitney test, U=9, n.s., non-significant, p=0.101. MP: ACSF, 39.5±2.8 m, n=6; PGB, 38.0±3.6 m, n=6; Mann–Whitney test, U=15, n.s., non-significant, p=0.675). (**F**) Experimental timeline. Right after the second injection of the acidic saline, one of the mouse groups was infused with PGB, and the other with ACSF. The behavior assessments were performed on day 14. (**G**) Top, representative travel trajectories of each ACSF and PGB group during the EPM test. Bottom, summary of the effects of PGB treatment on the open-arm time (ACSF, 22.1 ± 3.1%, n=8; PGB, 40.5 ± 4.8%, n=9; Mann–Whitney test, U=10, *p=0.011) and total distance (ACSF, 20.4±2.3, n=8; PGB, 25.2±3.1, n=9; Mann–Whitney test, U=25, n.s., non-significant, p=0.315). (**H**) Summary of the effects of PGB treatment on the number of buried marbles (ACSF, 17.6±1.2, n=5; PGB, 9.0±0.9, n=6; Mann–Whitney test, U=0, **p=0.004). (**I**) Summary of the effects of PGB treatment on the immobility time (ACSF, 54.8±0.9, n=5; PGB, 41.0±2.4, n=6; Mann–Whitney test, U=0, **p=0.004). (**J**) Top, representative travel trajectories of each ACSF and PGB group during the three-chamber sociability test. Bottom, summary of the effects of PGB treatment on social-zone time (ACSF, 33.2±2.6, n=8; PGB, 42.1±1.9, n=9; Mann–Whitney test, U=11, *p=0.015) and total distance (ACSF, 39.8±1.8, n=8; PGB, 40.7±2.8, n=9; Mann–Whitney test, U=33, n.s., non-significant, p=0.791).

The online version of this article includes the following source data and figure supplement(s) for figure 2:

**Source data 1.** Numerical data to support the graphs in *Figure 2*.

**Figure supplement 1.** Correlation of the effect of pregabalin (PGB) on the mechano-sensitivity and emotional behavior.

**Figure supplement 1—source data 1.** Numerical data to support the graphs in *Figure 2—figure supplement 1*.

immediately after the second acidic saline injection interfered with the hyperalgesic priming, thereby preventing the development of chronic pain-related comorbid affective emotions in the MP mice.

## MP was associated with enhanced synaptic transmission and neuronal excitability in the CeA-SST neurons

To characterize maladaptive changes in the amygdala circuits in chronic MP states, we recorded the excitatory synaptic transmission to the CeA-SST and CeA-PKCδ neurons using whole-cell patch-clamp recording in brain slices from the Ctrl and MP mice (*Figure 3A*). Using the Cre-loxP recombination approach, we identified the CeA-SST and CeA-PKCδ neurons in the SST-Cre;Ai14 and PKCδ-Cre;Ai14 mice, respectively (*Figure 3B*). Compared to the Ctrl mice, MP mice had an increase in the frequency of spontaneous excitatory postsynaptic currents (sEPSCs) in the CeA-SST neurons (*Figure 3C*). The amplitude of sEPSCs in the MP mice, however, was not significantly different from that in the Ctrl mice (*Figure 3C*). In contrast to the CeA-SST neurons, recordings from the CeA-PKCδ neurons of the MP mice showed a significant decrease in the sEPSC frequency; however, no change was observed in the sEPSC amplitude compared to the Ctrl mice (*Figure 3D*). Similar results were observed in miniature excitatory postsynaptic currents (mEPSCs; *Figure 3—figure supplement 1A, B*), suggesting that excitatory synaptic transmission onto the CeA-SST neurons was strengthened, while that onto the CeA-PKCδ neurons was weakened.

Next, we investigated whether the intrinsic excitability of the CeA-SST and CeA-PKCδ neurons was altered in the MP mice (*Figure 3E*). To determine the changes in intrinsic properties, the CeA neurons were recorded in the presence of synaptic blockers, which blocked glutamatergic and GABAergic transmission (see Materials and methods). Compared to the Ctrl mice, resting membrane potentials and input resistance of the CeA-SST neurons and CeA-PKCδ neurons in the MP mice were not significantly altered (*Figure 3—figure supplement 1C-F*). However, the CeA-SST neurons in the MP mice generated more spikes in response to prolonged current injections (*Figure 3E* left, *Figure 3F*). Conversely, the CeA-PKCδ neurons in the MP mice generated fewer spikes (*Figure 3E* right, *Figure 3G*). Consistent with these results, the CeA-SST neurons exhibited a lower rheobase (*Figure 3H*), while the CeA-PKCδ neurons exhibited a higher rheobase in the MP mice compared to the Ctrl mice (*Figure 3H*). These results indicated that cell-intrinsic mechanisms that induce enhancement of excitability in the MP mice are CeA-SST neuron-specific. Consistent with these findings, we observed that the number of phosphorylated extracellular signal-regulated kinase (pERK)-positive CeA cells was increased in the

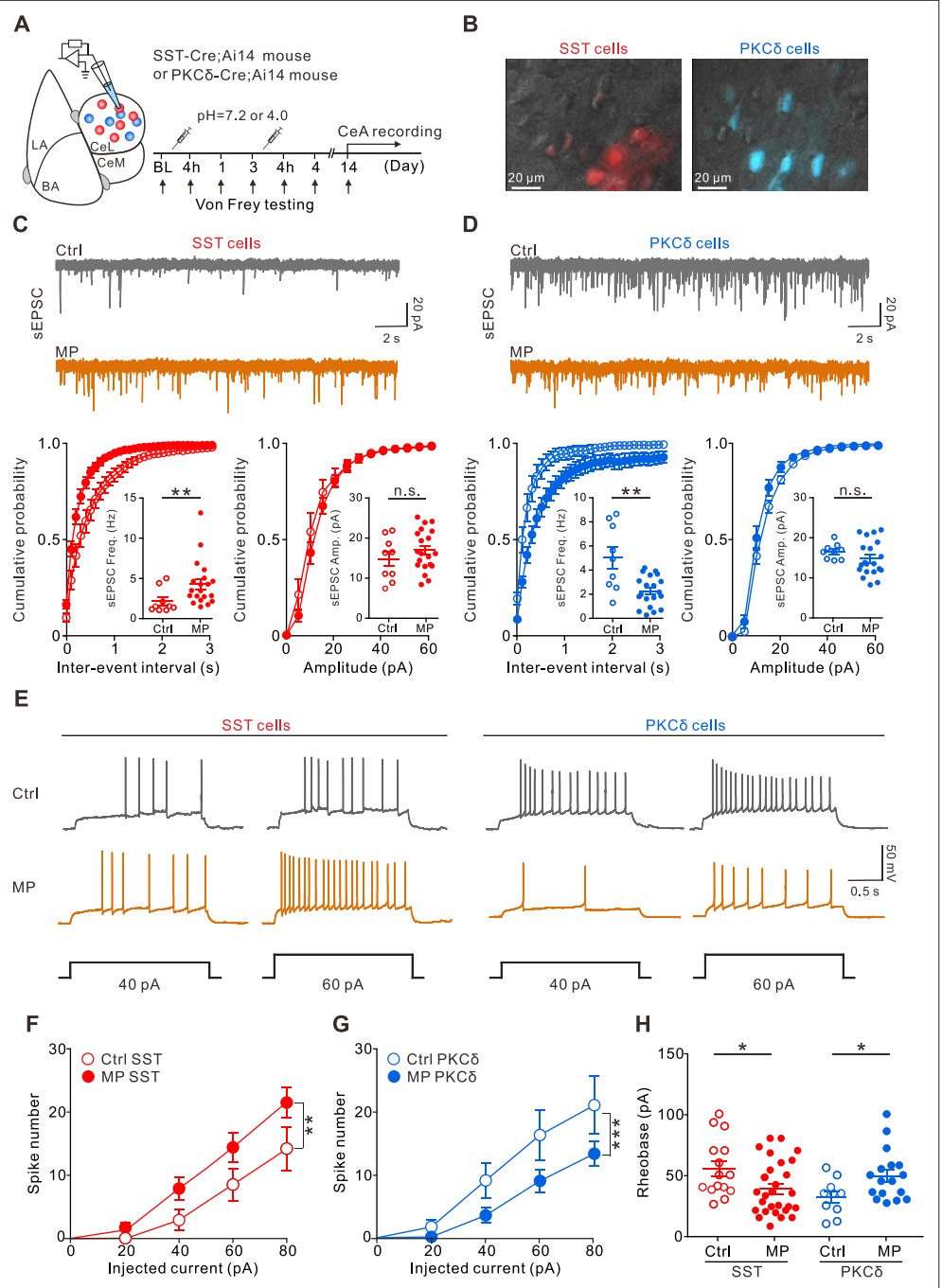

**Figure 3.** Altered excitatory transmission to the central nucleus of the amygdala (CeA) neurons and changed CeA neuron excitability in muscle pain (MP) mice. (**A**) Experimental schematic and timeline. (**B**) Overlay of epifluorescence and IR-DIC images showing somatostatin-expressing (SST) and protein kinase C delta-expressing (PKCδ) neurons in the CeA. Sections from the SST-Cre;Ai14 and PKCδ-Cre;Ai14 mouse brains. (**C**) Top, representative spontaneous excitatory postsynaptic current (sEPSC) traces recorded from the CeA-SST neurons of the Ctrl and MP mice. Bottom, cumulative probability of inter-event interval (Ctrl, n=9; MP, n=20; Kolmogorov–Smirnov test, **p=0.0014. Inset, summary of sEPSC frequency, Ctrl, 2.2±0.5 Hz, n=9; MP, 4.3±0.6 Hz, n=20; Mann–Whitney test, U=33, **p=0.006) and amplitude (Ctrl, n=9; MP, n=20; Kolmogorov–Smirnov test, n.s., non-significant, p=0.998. Inset, summary of sEPSC amplitude, Ctrl, 14.9±1.8 pA, n=9; MP, 16.9±1.1 pA, n=20; Mann–Whitney test, U=72, n.s., non-significant, p=0.404). (**D**) Top, representative sEPSC traces recorded from the CeA-PKCδ neurons of the Ctrl and MP mice. Bottom, cumulative probability of inter-event interval (Ctrl, n=9; MP, n=20; Kolmogorov–Smirnov test, ****p<0.0001. Inset, summary of sEPSC frequency, Ctrl, 5.1±0.9 Hz, n=9; MP, 2.3±0.3 Hz, n=20;

*Figure 3 continued on next page*

*Figure 3 continued*

Mann–Whitney test, U=36, **p=0.0097) and amplitude (Ctrl, n=9; MP, n=20; Kolmogorov–Smirnov test, n.s., non-significant, p=0.996. Inset, summary of sEPSC amplitude, Ctrl, 16.5±0.6 pA, n=9; MP, 14.7±1.0 pA, n=20, Mann–Whitney test, U=60, n.s., non-significant, p=0.167). (**E**) Representative responses of the CeA-SST and CeA-PKCδ neurons in the Ctrl and MP mice to depolarizing current injections. (**F**) Plot of number of spikes in the CeA-SST neurons against injected current (Ctrl, n=15; MP, n=28; two-way analysis of variance (ANOVA) with Bonferroni's multiple comparison test, F(1,202) = 10.49, **p=0.0014). (**G**) Plot of number of spikes in CeA-PKCδ neurons against injected current (Ctrl, n=11; MP, n=20; two-way ANOVA with Bonferroni's multiple comparison test, F(1,143) = 12.77, ***p=0.0005). (**H**) Summary of rheobase of CeA-SST neurons (Ctrl, 55.7±6.1 pA, n=15; MP, 38.7±4.1 pA, n=28; Mann–Whitney test, U=117, *p=0.017) and CeA-PKCδ neurons (Ctrl, 32±4.7 pA, n=10; MP, 49.6±5.0 pA, n=17; Mann–Whitney test, U=42.5, *p=0.032).

The online version of this article includes the following source data and figure supplement(s) for figure 3:

**Source data 1.** Numerical data to support the graphs in *Figure 3*.

**Figure supplement 1.** Comparison of miniature excitatory postsynaptic current (mEPSC), resting potential, and input resistance of the central nucleus of the amygdala (CeA) neurons in the Ctrl and muscle pain (MP) mice.

**Figure supplement 1—source data 1.** Numerical data to support the graphs in *Figure 3—figure supplement 1*.

**Figure supplement 2.** The total number of SST(+)pERK(+) cells in the central nucleus of the amygdala (CeA) was increased in muscle pain (MP) mice.

**Figure supplement 2—source data 1.** Numerical data to support the graphs in *Figure 3—figure supplement 2*.

**Figure supplement 3.** Histogram of the decay time.

**Figure supplement 4.** Altered synaptic transmission and excitability of different firing phenotypes of the somatostatin-expressing central nucleus of the amygdala (CeA-SST) and CeA-protein kinase C delta-expressing (PKCδ) neurons in the muscle pain (MP) mice.

**Figure supplement 4—source data 1.** Numerical data to support the graphs in *Figure 3—figure supplement 4*.

---

MP mice (*Figure 3—figure supplement 2A, B*). Moreover, increased pERK-positive cells were mostly CeA-SST cells (*Figure 3—figure supplement 2A, C, D*).

## Inhibition of the CeA-SST neurons alleviated the pain and affective symptoms

The increased CeA-SST excitability was associated with chronic pain phenotypes in the MP mice. We sought to determine the causal role of the CeA-SST neurons in the MP development. Therefore, we tested whether suppression of neuronal excitability of the CeA-SST neurons during the initial muscle injuries was sufficient to interfere with the development of chronic pain states. A Cre-dependent adeno-associated virus serotype 5 (AAV5) encoding an inhibitory designer receptor (hM4Di) was injected bilaterally into the CeA of SST-Cre mice (*Figure 4A*). Most (~80%) hM4Di-expressing cells, as identified by the mCherry expression, were immunoreactive for SST (*Figure 4B*; *Figure 4—figure supplement 1*). Whole-cell current-clamp recordings from acute amygdala slices prepared from the mice expressing the hM4Di in the CeA demonstrated that bath application of clozapine-N-oxide (CNO; 5 µM) inhibited spontaneous firing in the hM4Di-expressing SST neurons (*Figure 4C*1). In contrast, CNO had no effect on the firing of the SST neurons, which only expressed mCherry (*Figure 4C*2). To test the essential role of CeA-SST neuronal activity in MP induction, CNO (5 mg/kg body weight) was intraperitoneally (i.p.) injected into the mice immediately after the second acidic saline injection. Silencing the CeA-SST neurons during the primed state prevented the development of MP in the hM4Di-expressing mice compared to the mCherry-expressing mice (*Figure 4D*). This result suggests that activation of the CeA-SST neurons following peripheral injury was necessary for pain chronification.

Next, we tested the causal role of the CeA-SST neurons in MP-associated behaviors. CNO was i.p. injected into the mice following the establishment of MP (*Figure 4E*). We found that inhibiting the CeA-SST neurons in the MP mice 2 weeks following MP induction was still effective in increasing the PW threshold on day 15 (*Figure 4E*), and the pain-reduction effect lasted for 24 hr and persisted if CNO was administered repeatedly (*Figure 4—figure supplement 2*). On the same day following the CNO injection, we also tested the effect of inhibiting the CeA-SST neurons on the affective symptoms. Compared to the mCherry-MP mice, the hM4Di-MP mice spent more time in the open arms during

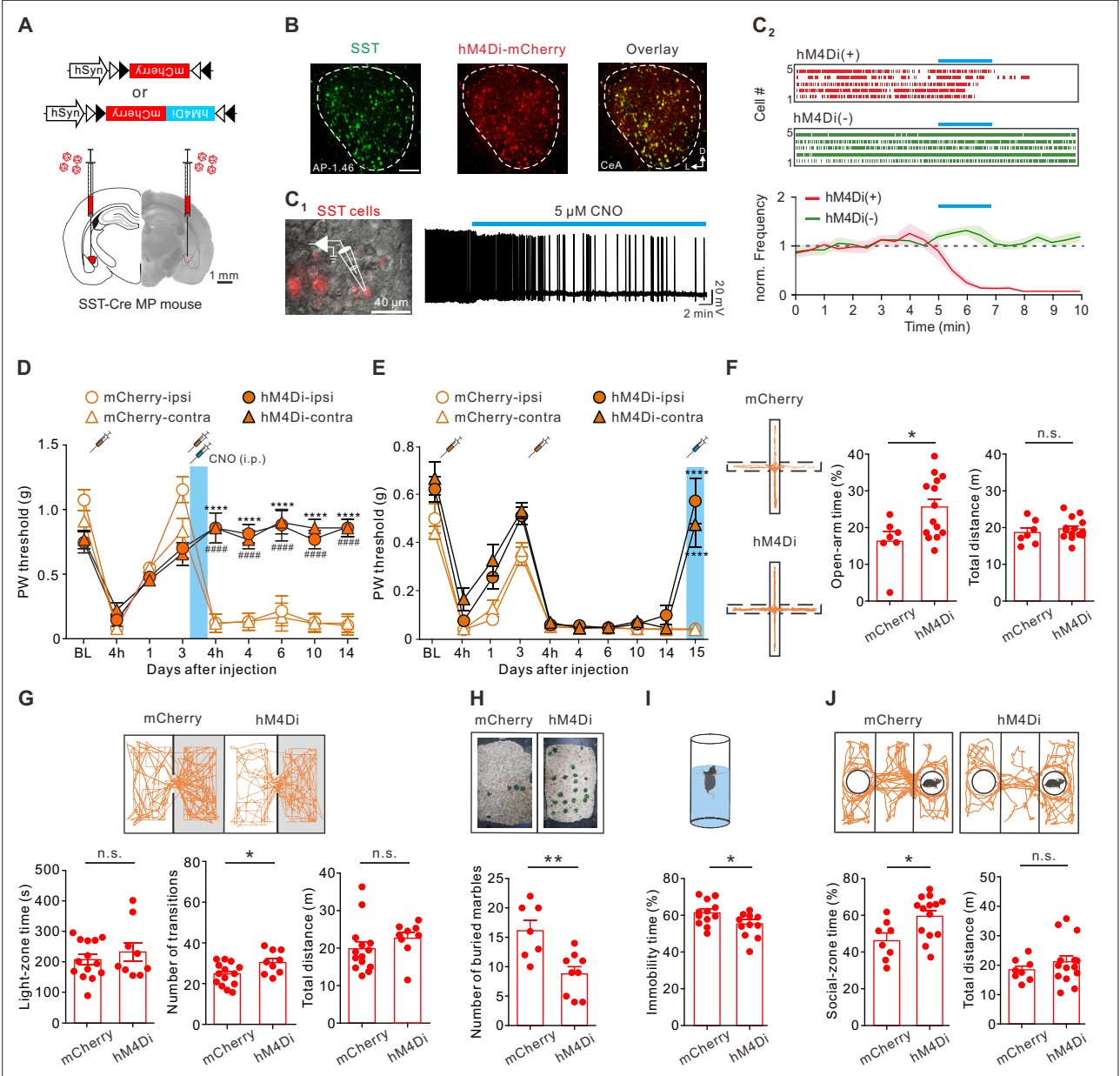

**Figure 4.** Inactivating the somatostatin-expressing central nucleus of the amygdala (CeA-SST) neurons alleviated pain and affective symptoms. (**A**) Viral constructs and experimental schematic. (**B**) Left, representative image of the CeA-SST neurons (green) recognized by an antibody. Middle, representative image of expression of hM4Di-mCherry (red) virus. Right, overlapping image of hM4Di-mCherry and SST antibody signals. (**C₁**) Left, overlay of epifluorescence and IR-DIC images showing mCherry(+) neurons in the CeA. Right, membrane firing frequency changes of an hM4Di-expressing CeA-SST neuron before and after bath application of clozapine-N-oxide (CNO). (**C₂**) Top, representative of raster plots of hM4Di(+) and hM4Di(-) neurons. Blue bar indicates 5 μM CNO application for 2 min. Bottom, normalized firing frequency in both hM4Di(+) and hM4Di(-) neurons (hM4Di(-), n=10; hM4Di(+), n=10; two-way analysis of variance (ANOVA) with Bonferroni's multiple comparison test, $F_{(1,376)}$ = 231.1, ****p<0.0001). (**D**) Effects of the CNO treatment on day 3 on the paw withdrawal (PW) threshold (mCherry, n=5; hM4Di, n=9; two-way ANOVA with Tukey's post hoc test, $F_{(3,216)}$ = 46.82, ****,####p<0.0001; * indicates the comparison between the ipsilateral hindpaw; # indicates the comparison between the contralateral hindpaws. The blue area indicates the period of CNO treatment). (**E**) Effects of the CNO treatment on the PW threshold in the muscle pain (MP) mice (mCherry, n=10; hM4Di, n=9; two-way ANOVA with Tukey's post hoc test, $F_{(9,340)}$ = 124.1, ****p<0.0001 relative to day 14. The blue area indicates the period of CNO treatment). (**F**) Left, representative trajectories of each MP group during the elevated plus maze (EPM) test. Right, summary of the effects of CNO treatment on the open-arm time (mCherry, 16.2 ± 2.5%, n=7; hM4Di, 25.5 ± 2.2%, n=14; Mann–Whitney test, U=19, *p=0.025) and total distance (mCherry, 18.7±1.3 m, n=7; hM4Di, 19.7±0.8 m, n=14; Mann–Whitney test, U=37, n.s., non-significant, p=0.383). (**G**) Top, representative travel paths of each MP group during the light/dark (L/D) box test. Bottom, summary of the CNO effect on the light-zone time (mCherry, 204.9±16.7 s, n=14; hM4Di, 230.2±30.3 s, n=9; Mann–Whitney test, U=57, n.s., non-significant, p=0.73), number of transitions (mCherry, 23.6±1.5, n=14; hM4Di, 30.0±1.9,

*Figure 4 continued on next page*

*Figure 4 continued*

n=9; Mann–Whitney test, U=28.5, *p=0.028) and total distance (mCherry, 20.0±1.8 m, n=14; hM4Di, 22.7±1.6 m, n=9; Mann–Whitney test, U=34, n.s., non-significant, p=0.072). (**H**) Top, representative images of the marble burying test. Bottom, summary of the effects of CNO treatment on the number of buried marbles (mCherry, 16.1±1.8, n=7; hM4Di, 8.8±1.2, n=9; Mann–Whitney test, U=6.5, **p=0.005). (**I**) Top, schematic of the forced swim test (FST) setup. Bottom, summary of relative time of immobility (mCherry, 61.4 ± 1.9%, n=12; hM4Di, 54.2 ± 2.0%, n=11; Mann–Whitney test, U=29, *p=0.022). (**J**) Top, representative travel paths of each MP group during the three-chamber sociability test. Bottom, summary of the effects of CNO treatment on the social-zone time (mCherry, 46.7 ± 3.7%, n=8; hM4Di, 60.0 ± 3.1%, n=14; Mann–Whitney test, U=22, *p=0.019) and total distance (mCherry, 18.5±1.3 m, n=8; hM4Di, 21.2±2.1 m, n=14; Mann–Whitney test, U=47, n.s., non-significant, p=0.55).

The online version of this article includes the following source data and figure supplement(s) for figure 4:

**Source data 1.** Numerical data to support the graphs in *Figure 4*.

**Figure supplement 1.** Colocalization of specific somatostatin-expressing (SST) biomarker with virus expression.

**Figure supplement 1—source data 1.** Numerical data to support the graphs in *Figure 4—figure supplement 1*.

**Figure supplement 2.** Suppression of the somatostatin-expressing central nucleus of the amygdala (CeA-SST) neurons increased the withdrawal threshold in the muscle pain (MP) mice.

**Figure supplement 2—source data 1.** Numerical data to support the graphs in *Figure 4—figure supplement 2*.

**Figure supplement 3.** Suppressing somatostatin-expressing central nucleus of the amygdala (CeA-SST) neuron excitability in the Ctrl mice exerted little effect on the nociception and affective behaviors.

**Figure supplement 3—source data 1.** Numerical data to support the graphs in *Figure 4—figure supplement 3*.

**Figure supplement 4.** Enhancing central nucleus of the amygdala (CeA)-protein kinase C delta-expressing (PKCδ) neuron excitability reduced pain, but failed to alleviate affective symptoms.

**Figure supplement 4—source data 1.** Numerical data to support the graphs in *Figure 4—figure supplement 4*.

**Figure supplement 5.** Proposed wiring diagram of central nucleus of the amygdala (CeA) circuits, extrinsic pathways, and behavioral responses.

the EPM test (*Figure 4F*), exhibited increased transitions between light and dark zones in the L/D box test (*Figure 4G*), buried fewer marbles in the marble burying test (*Figure 4H*), and immobilized less in the FST (*Figure 4I*). Moreover, the hM4Di-MP mice spent more time in the social zone in the three-chamber sociability test (*Figure 4J*). However, the total travel distances of the mCherry-MP and hM4Di-MP mice were not significantly different (*Figure 4F, G and J*). Notably, chemogenetic inhibition of the CeA-SST neurons in the hM4Di-Ctrl mice had little effects on the PW threshold and on their respective performances in the EPM, L/D box, marble burying test, FST, and sociability test (*Figure 4—figure supplement 3*). Collectively, these observations suggest that inactivating the CeA-SST neurons selectively reduced pain and alleviated the anxiety- and depression-like behaviors in the MP mice.

Both enhanced CeA-SST and decreased CeA-PKCδ neuronal activity were observed in the MP mice. These two types of neurons reciprocally inhibit each other. We subsequently tested whether increasing CeA-PKCδ neuronal activity could also reduce mechanical hyperalgesia and anxiety-like behaviors. The excitatory designer receptor (hM3Dq) was virally expressed in the CeA-PKCδ neurons (*Figure 4—figure supplement 4A*). Most of the hM3Dq-expressing cells, as identified by the mCherry expression, were immunoreactive for PKCδ (*Figure 4—figure supplement 4B*). Whole-cell current-clamp recordings demonstrated that bath application of CNO (5 μM) selectively enhanced spontaneous firing in the hM3Dq-expressing CeA-PKCδ neurons as compared to the mCherry-expressing CeA-PKCδ neurons (*Figure 4—figure supplement 4C*). Like the silencing of the hM4Di-expressing CeA-SST neurons, the activation of the hM3Dq-expressing CeA-PKCδ neurons greatly increased the PW threshold in the MP mice (*Figure 4—figure supplement 4D*). Intriguingly, unlike the results of inactivating the CeA-SST neurons, enhancing the CeA-PKCδ neurons had little effect on the anxiety-like behaviors. There were no changes in the open-arm time (*Figure 4—figure supplement 4E*), number of transitions during the L/D box test (*Figure 4—figure supplement 4F*), and number of buried marbles (*Figure 4—figure supplement 4G*). Similarly, enhancing the CeA-PKCδ neurons did not improve depression-like behavior (*Figure 4—figure supplement 4H*) and sociability (*Figure 4—figure supplement 4I*). Taken together, these results suggest that activation of the CeA-PKCδ neurons is sufficient to reduce mechanical hyperalgesia in the MP mice. However, changes in the excitability of the CeA-PKCδ neurons were not causally related to MP-related affective behaviors.

## PGB suppressed the CeA-SST neuron excitability and nociceptive transmission onto the CeA-SST neurons

Since enhanced CeA-SST neuronal activity was observed in the MP mice, we tested the effect of PGB on excitatory synaptic transmission onto the CeA-SST neurons. To this end, we injected an AAV5-CaMKIIα-ChR2-eYFP virus into the PBN of the SST-Cre;Ai14 mice. Four weeks later, these mice were randomly included as either the Ctrl or MP mice (*Figure 5A*). Using the optogenetic stimulation, we recorded light-evoked EPSCs from the CeA-SST neurons of the Ctrl or MP mice in response to short light (470 nm, 5 ms) illumination to PBN terminals in the CeA region (*Figure 5B*; Ctrl, left; MP, right). Bath application of PGB (500 µM) decreased the amplitude of light-evoked EPSCs in some CeA-SST neurons of both the Ctrl and MP mice (*Figure 5B and D*). The percentage of neurons that were sensitive to PGB inhibition was increased in the MP mice (*Figure 5C*, left), although the degree of inhibition between these two groups was similar (*Figure 5C*, right). Consistent with the presynaptic effect of PGB on α2δ VGCC subunits, the reduction of EPSCs was concomitant with an increase in the paired-pulse ratio (the interpulse interval = 200 ms; *Figure 5E*). Similar effects of PGB on synaptic transmission were found at the PBN to CeA-PKCδ neuron synapses (*Figure 5—figure supplement 1*).

Finally, we tested whether intraperitoneal injection of PGB suppresses the CeA-SST neuron activity using in vivo calcium imaging. To selectively detect the calcium activities of the CeA-SST neurons, the SST-Cre mice were virally transduced to express the calcium indicator GCaMP6s and implanted with the GRIN lens above the CeA for 4 weeks before the test. Calcium activities of single CeA-SST neurons were detected before and 45 min following intraperitoneal injection of PGB (30 mg/kg; *Figure 5F*), which is known to significantly increase the mechanical threshold of the hindpaws in this MP model (*Yokoyama et al., 2007*). After intraperitoneal injection of PGB, the CeA-SST neurons exhibited decreased calcium event frequency (*Figure 5G and H*, left) and mean amplitude (ΔF/F; *Figure 5G and H*, right). The analysis of the area under the curve (AUC) of spontaneous calcium activities suggested that approximately 56% (26/46 cells) of CeA-SST neurons were inhibited, 11% (5/46 cells) of CeA-SST neurons were excited, and 33% of (15/46 cells) CeA-SST neurons were insensitive to PGB application (*Figure 5I and J*). Taken together, these results suggest that PGB suppressed glutamate transmission onto the CeA-SST neurons, thereby reducing mechanical allodynia and reversed anxiety- and depression-like behaviors in the MP mice.

## Discussion

In this study, we explored the role of the CeA neurons in central sensitization of chronic pain using the acid-induced MP model. This model developed by *Sluka et al., 2001* is a generally accepted preclinical animal model for chronic musculoskeletal pain syndrome. The second acidic saline injection during the hyperalgesic priming promoted pain chronification. Local PGB application into the CeA reduced mechanical allodynia and reversed affective behaviors in the MP mice. Intriguingly, intra-CeA PGB application or selective inactivation of CeA-SST neurons immediately following the second acidic saline injection prevented the chronification of pain. Notably, the MP was accompanied by enhanced glutamatergic transmission onto the CeA-SST neurons and decreased synaptic transmission onto the CeA-PKCδ neurons. Furthermore, the CeA-SST neuron excitability and CeA-PKCδ neuron excitability was increased and decreased, respectively. In agreement with the role of CeA-SST neurons in central sensitization, chemogenetic inactivation or pharmacological suppression of the CeA-SST neurons by PGB effectively alleviated MP and comorbid affective behaviors.

### Effect of PGB on pain and emotional behaviors

The effect of PGB on the PW threshold could be replicated by either reducing or elevating the CeA-SST and CeA-PKCδ neuron activities, respectively; however, this is not the case for the anxiety-like affective behaviors. Therefore, the effect of PGB on the PW threshold and emotional behaviors in the MP mouse model may be caused by distinct simultaneously occurring circuit mechanisms. To further elucidate this, we plotted the changes of the immobility time in the FST or number of buried marbles in the marble burying test against the change of mechano-sensitivity to examine the effect of PGB on pain sensitivity and emotional behaviors. Only the data obtained from the FST and marble burying test were used to examine the effect of PGB on the pain sensitivity and emotional behaviors because these two behavioral assays could be tested before and after PGB application on the same animals.

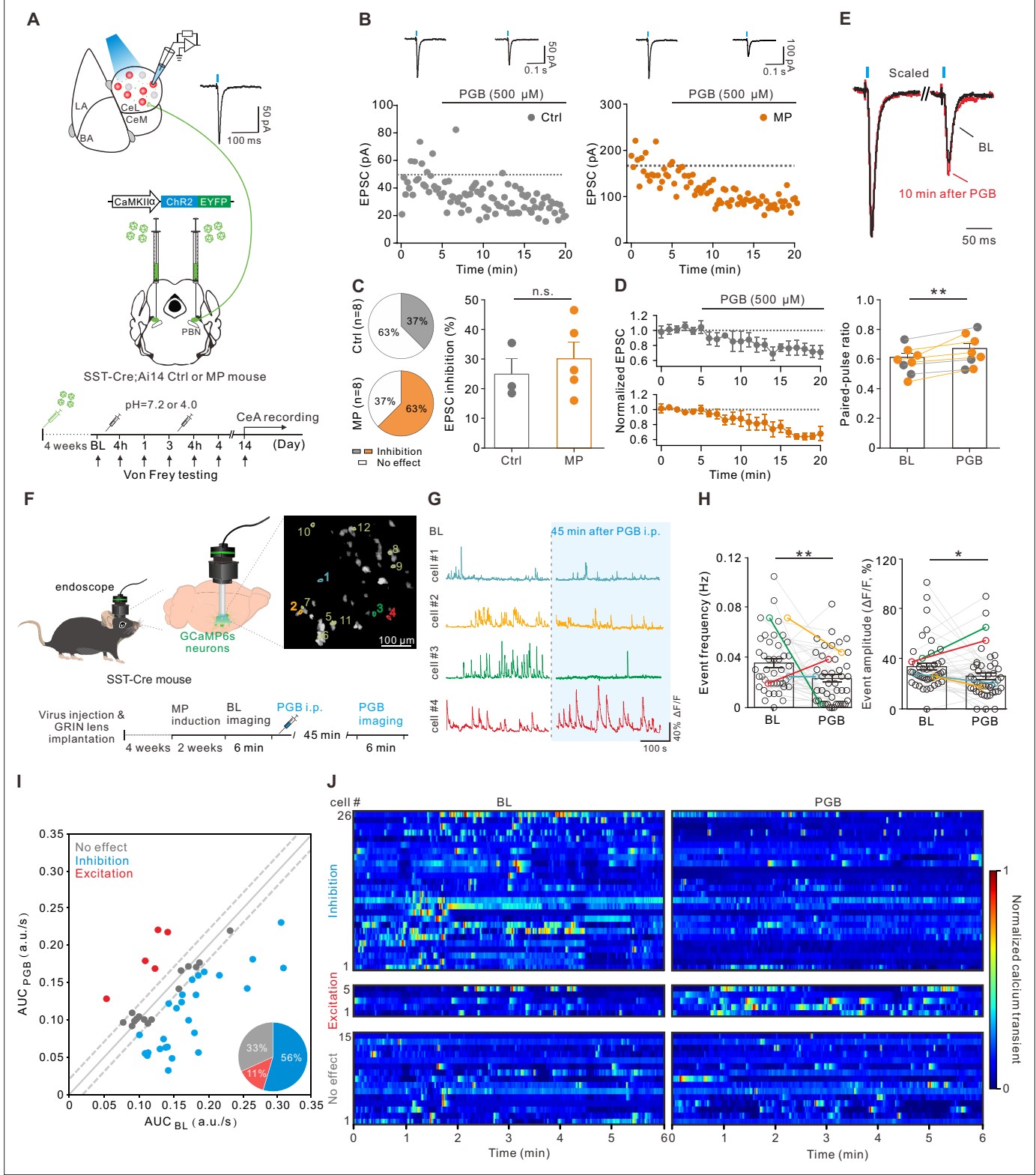

**Figure 5.** Pregabalin (PGB) suppressed the somatostatin-expressing central nucleus of the amygdala (CeA-SST) neuron excitability and glutamate release from the parabrachial nucleus (PBN) to the CeA-SST neurons. (**A**) Experimental schematic and timeline. (**B**) Left, amplitude of the light-evoked excitatory postsynaptic currents (EPSCs) before and after PGB application of CeA-SST neurons in a representative Ctrl mouse. Right, amplitude of the light-evoked EPSCs before and after PGB application of the CeA-SST neurons in a representative muscle pain (MP) mouse; traces of EPSCs of the cell are shown above. (**C**) Left, pie chart showing the percentage of inhibition neurons after PGB treatment (Ctrl, 37%, n=8 cells from 5 mice; MP,

*Figure 5 continued on next page*

*Figure 5 continued*

63%, n=8 cells from 4 mice; Chi-square test, p=0.317). Right, percentage of EPSC inhibition of PGB-sensitive neurons in the Ctrl and MP groups (Ctrl, 24.9 ± 5.3%, n=3; MP, 30.1 ± 5.5%, n=5; Mann–Whitney test, U=5, n.s., non-significant, p=0.5). (**D**) Cells with EPSC inhibition >×1 $SD_{BL}$ are pooled and plotted for the normalized EPSCs over time (Ctrl, n=3; MP, n=5). (**E**) Top, representative EPSC traces before (black) and 10 min after PGB application (red). Average trace after PGB application is normalized to the peak of average EPSC trace during baseline (BL). Paired-pulse interval = 200 ms. Bottom, summary of paired-pulse ratio (PPR) of cells shown in (**D**) before and after PGB application (BL, 0.58±0.03, n=8; PGB, 0.65±0.04, n=8; gray circle: Ctrl group; orange circle: MP group; Wilcoxon matched-pairs signed rank test, **p=0.008). (**F**) Left and bottom, experimental schematic and timeline. Right, representative image of CeA-SST neurons labeled with GCaMP6s. Cells with their calcium traces shown in the panel G were labeled. (**G**) Representative traces from cells of interest (color-matched in the panel F right). (**H**) The frequency of calcium events for CeA-SST neurons before and after PGB i.p. injection (BL, 0.036±0.003 Hz; PGB, 0.023±0.003 Hz, n=46 cells from three mice; Wilcoxon matched-pairs signed rank test, **p=0.004) and the percentage of ΔF/F before and after PGB i.p. injection (BL, 34.1 ± 2.8%; PGB, 26.4 ± 2.7%, n=46 cells from three mice; Wilcoxon matched-pairs signed rank test, *p=0.045). (**I**) The area under the curve (AUC) of calcium traces of CeA-SST neurons before and after PGB application. Neurons with ΔAUC < $1 \times \sigma_{dev}$(0.02 a.u./s) were distributed within two dashed lines. Pie chart, the percentage of neuronal responses after PGB treatment (n=46 cells). (**J**) The heatmaps of normalized calcium activities before and after the PGB treatment. Neurons were grouped based on the classification in (**I**).

The online version of this article includes the following source data, source code, and figure supplement(s) for figure 5:

**Source code 1.** Related to *Figure 5I and J*.

**Source data 1.** Numerical data to support the graphs in *Figure 5*.

**Figure supplement 1.** Effect of pregabalin (PGB) on neurotransmission at the parabrachial nucleus (PBN) to the central nucleus of the amygdala (CeA)-protein kinase C delta-expressing (PKCδ) neuron synapses.

**Figure supplement 1—source data 1.** Numerical data to support the graphs in *Figure 5—figure supplement 1*.

**Figure supplement 2.** Comparison of light-evoked excitatory postsynaptic current (EPSC) amplitude and paired-pulse ratio (PPR) at the parabrachial nucleus (PBN) to the central nucleus of the amygdala (CeA) neuron synapses.

**Figure supplement 2—source data 1.** Numerical data to support the graphs in *Figure 5—figure supplement 2*.

After PGB application, no positive correlation was observed between the increase in the PW threshold and the reduction of the immobility time. On the other hand, only a weak positive correlation between the change of PW threshold and change in the number of buried marbles was detected (*Figure 2—figure supplement 1*). These results indicate that PGB may alleviate the mechano-sensitivity and emotional behavior through different subtypes of the CeA-SST neurons or distinct circuit mechanisms (*Figure 4—figure supplement 5*; see Discussion).

## Maladaptive rewiring of glutamatergic synapses onto the CeA neurons

Synaptic modifications and changes of intrinsic excitability play a crucial role in health and disease (*Bliss et al., 2014*; *Fu et al., 2008*; *Ikeda et al., 2007*; *Kuner and Flor, 2016*; *Penzo et al., 2015*). The formation of long-term potentiation at the synapses in pain-related brain regions is considered one of the synaptic mechanisms underlying the transition of acute to chronic MP (*Min et al., 2011*). The protein kinase C (PKC)-extracellular signal-regulated kinase (ERK) pathway plays a key role in the maladaptive changes in the neural circuitry of the MP mice (*Chen et al., 2010*; *Cheng et al., 2011*; *Min et al., 2011*). For instance, MP is associated with enhanced pERK expression in the paraventricular thalamus (PVT) and CeA (*Chen et al., 2010*; *Cheng et al., 2011*; *Figure 3—figure supplement 2*). The PVT, a structure that is readily activated by both physical and psychological stressors, projects to the CeA (*Penzo et al., 2015*). Several studies supported the involvement of the PKC and/or ERK activity in these two brain regions. First, inhibition of the PKC or ERK activity in the PVT, a major input to the CeA, prevents the chronification of pain (*Chen et al., 2010*). Second, upregulation of α-amino-3-hydroxy-5-methyl-4-isoxazolepropionic acid receptors function at the PBN-capsular central amygdala neurons (CeAC) synapse in the MP mice requires the activation of the PKC-mitogen-activated protein kinase (MEK)-ERK pathway in the CeAC neurons (*Cheng et al., 2011*). Finally, PKC-dependent, but ERK-independent, enhancement of synaptic release occurs at the PBN-CeAC synapses in the MP mice (*Cheng et al., 2011*).

Here, we reported an increase in the sEPSC and mEPSC frequencies but not their amplitudes in the CeA-SST neurons. The EPSCs and PPR at the PBN-CeA-SST neuron synapses were not significantly altered in the MP mice compared to the Ctrl mice (*Figure 5—figure supplement 2*), suggesting other possible inputs, which may contribute to increased spontaneous synaptic events. Several excitatory inputs impinge onto CeA neurons (*Fadok et al., 2018*). The CeA receives nociceptive inputs from the dorsal horn via the PBN and affect-related information from the basolateral amygdala (BLA) (*Ikeda*

*et al., 2007*; *Kuner and Kuner, 2021*; *Li and Sheets, 2020*; *Wilson et al., 2019*). Additionally, the PVT, which is a structure that is readily activated by both physical and psychological stressors, projects onto the CeA (*Penzo et al., 2015*). The sEPSC decay time analysis revealed at least two populations of sEPSCs, suggesting heterogeneous excitatory inputs impinging on different spatial domains of the CeA neuron dendrites (*Figure 3—figure supplement 3*). Previous studies have reported that synaptic inputs from the PVT and BLA onto the CeA are strengthened in chronic pain conditions (*Ikeda et al., 2007*; *Liang et al., 2020*). Considering the role of ERK in the PVT for MP (*Chen et al., 2010*), the PVT input most likely contributes to the enhancement of glutamate release to the CeA-SST neurons in the MP mice. Moreover, we cannot exclude the contribution of the BLA input, because BLA-CeA synapses are potentiated in chronic neuropathic pain (*Ikeda et al., 2007*).

In addition to synaptic plasticity, ERK is probably involved in the cell-intrinsic mechanisms. Considering that pERK is increased in CeA-SST neurons in the MP mice (*Figure 3—figure supplement 2*), we speculate that the ERK activity may contribute to the increase in intrinsic neuronal excitability of the CeA-SST neurons in the MP mice. The CaMKII-PKA/PKC-ERK-cAMP response element binding protein (CREB) pathway triggers gene transcription and protein synthesis (*Kolch, 2000*). The activated ERK1/2 can undergo phosphorylation after translocating to the nucleus and then activate the downstream transcriptional factor CREB (*Iyengar et al., 2017*). Activation of CREB triggers gene transcription and plasticity including regulation of a variety of ion channels, thereby changing the AP threshold and neuronal excitability (*Dong et al., 2006*; *Zhou et al., 2009*). For instance, downregulation of A-type $K^+$ channels such as $K_v4.2$ expression or upregulation of the voltage-gated Na channels such as $Na_v1.3$ and $Na_v1.7$ expressions can either decrease the rheobase or increase action potential numbers (*Bennett et al., 2019*; *Hu and Gereau, 2003*).

## Subtype of sensitized CeA neurons depends on the mouse model of chronic pain

A recent study by *Wilson et al., 2019*, demonstrated that neuropathic pain-induced activated forms of ERK (pERK) and cFos were preferentially expressed in the CeA-PKCδ neurons, which exhibited enhanced excitability relative to the CeA-SST neurons. Furthermore, chemogenetic silencing of CeA-PKCδ neurons or activation of the CeA-SST neurons reversed nerve-injury-induced hyperalgesia (*Wilson et al., 2019*). Similarly, another study demonstrated that chemogenetic inhibition of the CeA-PKCδ neurons reduced mechanical hyperalgesia in mice with formalin-induced pain (*Chen et al., 2022*). However, two recent studies contradicted this notion (*Hua et al., 2020*; *Zhou et al., 2019*). Using the spared nerve injury model, Zhou and colleagues demonstrated that a small population of CeA-SST neurons, which are glutamatergic and project to the lateral habenula (LHb), exhibited enhanced excitability and were involved in a neural circuit for hyperalgesia and comorbid depressive symptoms (*Zhou et al., 2019*). Moreover, a subpopulation of neurons co-expressing PKCδ and enkephalin confined to the mid-posterior axis of the CeA was shown to mediate the antinociceptive function in response to general anesthetics (*Hua et al., 2020*). In the present study, using the acid-induced MP model, we observed that chemogenetic inactivation of the CeA-SST neurons effectively alleviated MP and comorbid affective behaviors. Overall, the subtype of sensitized neurons in the CeA may vary based on the chronic pain models.

## The CeA contains functionally distinct neuronal populations

CeA-SST neurons form reciprocal connections with CeA-PKCδ neurons (*Ciocchi et al., 2010*; *Haubensak et al., 2010*). Notably, silencing the CeA-SST neurons in the MP model improved affective behaviors, while activating the CeA-PKCδ neurons had little effect on them. Our results support a notion that the CeA-SST neurons contain heterogeneous subpopulations (*Penzo et al., 2014*). The CeA-SST neurons could comprise at least two distinct subtypes of pro-nociceptive neurons based on their intrinsic firing properties and downstream targets in the MP model. A subset of the CeA-SST neurons could project onto the bed nucleus of the stria terminalis for approach-avoidance behavior, the midbrain ventrolateral periaqueductal gray for active-avoidance behavior, and the LHb for depression-like behavior. We speculate that the CeA-PKCδ neurons could inhibit the other subset of the CeA-SST neurons, which form reciprocal inhibition with the CeA-PKCδ neurons. In agreement with this hypothesis, chemogenetic activation of CeA-PKCδ neurons reduced tactile allodynia without influencing affective behaviors in MP. Notably, investigations on pain-related affective comorbidities

at the cellular level between these two neurons have not been reported in the acid-induced MP model. Nevertheless, the neural mechanisms by which the CeA-SST neurons and their downstream targets regulate negative affect have been characterized (*Ahrens et al., 2018*; *Haubensak et al., 2010*; *Li et al., 2013*; *Zhou et al., 2019*). Therefore, the hyperexcitability of the CeA-SST neurons and the resulting affective symptoms can be well explained by the downstream pathways of the CeA-SST neurons (*Figure 4—figure supplement 5*).

A potential caveat of this study is that single molecular markers such as SST or PKCδ may not be specific enough to identify functionally distinct CeA neuronal populations in the modulation of pain. For instance, these neurons display different intrinsic properties such as firing phenotypes (*Hou et al., 2016*). According to our previous study (*Hou et al., 2016*), we analyzed the CeA neurons shown in *Figure 3* and classified them into early-spiking (ES) and late-spiking (LS) neurons based on their spike delay at rheobase. Using the same criteria, the CeA neurons with a spike delay greater than 1.5 s were classified as LS neurons, whereas the rest were classified as ES neurons. The ratio of these two subpopulations (ES vs. LS subtype) in either CeA-SST or CeA-PKCδ neurons was not significantly altered in the MP mice (*Figure 3—figure supplement 4*). However, in the MP mice, the enhanced excitatory transmission was specific to the CeA-SST-ES neurons, whereas the depressed excitatory transmission was specific to the CeA-PKCδ-LS neurons. On the other hand, the modification of cell excitability in the MP mice was specific to ES neurons in both the CeA-SST and CeA-PKCδ neurons (*Figure 3—figure supplement 4*).

To achieve greater specificity, composite molecular markers and/or anatomical locations should be considered. A single molecularly defined population can be further divided based on its location within a brain area (*Li and Sheets, 2020*; *Wilson et al., 2019*). For example, spared nerve injury distinctly alters inputs from the PBN to the SST neurons based on their location in the lateral division of the CeA (CeL). Notably, the input from the PBN to the SST neurons in the capsular division of the CeA (CeC) is depressed, whereas the same input to the SST neurons in the CeL and the medial division of the CeA (CeM) is not altered. Moreover, our recent study (*Hou et al., 2016*) reported that the CeA-SST neurons exhibit a high degree of variation in the spike delay in response to the current injection in the slice recording and added an additional layer of heterogeneity. Considering the distinct synaptic and cellular properties of these subpopulations, they are likely to react differently to chemogenetic manipulations. Thus, while the entire molecularly defined neuronal population is targeted for manipulations, the net effects are likely to be dominated by the subpopulations that are preferentially expressed with optogenetic or chemogenetic actuators. Thus, a novel toolkit integrating anatomical, physiological, and molecular profiles of single neurons is warranted for functional dissection of CeA microcircuitry.

# Materials and methods

## Key resources table

| Reagent type (species) or resource | Designation | Source or reference | Identifiers | Additional information |
|---|---|---|---|---|
| Strain, strain background (mouse, male and female) | *Sst* $^{tm2.1(cre)Zjh}$/J | Jackson Laboratory | RRID: IMSR_JAX: 013044 | C57BL/6J genetic background |
| Strain, strain background (mouse, male and female) | Tg(Prkcd-glc-1/CFP,-cre) EH124Gsat/Mmucd | Mutant Mouse Resource and Research Center | RRID: MMRRC_011559-UCD | C57BL/6J genetic background |
| Strain, strain background (mouse, male and female) | B6.Cg-Gt(ROSA)26Sor$^{tm14 (CAG-tdTomato)}$ $^{Hze}$/J | Jackson Laboratory | RRID: IMSR_JAX: 007914 | C57BL/6J genetic background |
| Strain, strain background (mouse, male and female) | C57BL/6JNarl | National Laboratory Animal Center (Taiwan) | RMRC11005 | |
| Antibody | Rabbit anti-RFP (rabbit polyclonal) | Rockland | Cat#: 600-401-379S; RRID: AB_11182807 | 1:300 |
| Antibody | Rabbit anti-PKCδ (rabbit monoclonal) | Abcam | Cat#: ab182126; RRID:AB_2892154 | 1:500 |

*Continued on next page*

*Continued*

| Reagent type (species) or resource | Designation | Source or reference | Identifiers | Additional information |
|---|---|---|---|---|
| Antibody | Rat anti-SST (rat monoclonal) | Abcam | Cat#: ab30788; RRID:AB_778010 | 1:500 |
| Antibody | Goat anti-rabbit Alexa Fluor 488 (rabbit polyclonal) | Thermo Fisher Scientific | Cat#: A27034; RRID:AB_2536097 | 1:500 |
| Antibody | Goat anti-rabbit Alexa Fluor 594 (rabbit polyclonal) | Thermo Fisher Scientific | Cat#: A-11012; RRID:AB_2534079 | 1:500 |
| Antibody | Goat anti-rat Alexa Fluor 488 (rat polyclonal) | Thermo Fisher Scientific | Cat#: A-11006; RRID:AB_2534074 | 1:500 |
| Recombinant DNA reagent | AAV5-hSyn-DIO-mCherry | Addgene | RRID: Addgene_50459 | |
| Recombinant DNA reagent | AAV5-hSyn-DIO- hM3Dq-mCherry | Addgene | RRID: Addgene_44361 | |
| Recombinant DNA reagent | AAV5-hSyn-DIO-hM4Di-mCherry | Addgene | RRID: Addgene_44362 | |
| Recombinant DNA reagent | AAV5-CaMKIIα-ChR2-eYFP | University of North Carolina Vector Core | N/A | |
| Recombinant DNA reagent | AAV5-Syn-Flex-GCaMP6s-WPRE-SV40 | Addgene | RRID: Addgene_100845 | |
| Chemical compound, drug | MES hydrate | MERCK | M2933 | |
| Chemical compound, drug | Clozapine-N-oxide | MERCK | C0832 | |
| Chemical compound, drug | SR 95531 | Abcam | Ab120042 | |
| Chemical compound, drug | CGP 55845 | Tocris | 1248 | |
| Chemical compound, drug | Kynurenic acid | MERCK | K3375 | |
| Chemical compound, drug | Tetrodotoxin citrate | Tocris | 1069 | |
| Chemical compound, drug | Pregabalin | Pfizer | N/A | |
| Software, algorithm | Tru-scan 2.0 system | Coulbourn instruments | N/A | https://www.coulbourn.com/category_s/262.htm |
| Software, algorithm | EthoVision XT 13 | Noldus Information Technology | RRID:SCR_000441 | https://www.noldus.com/ethovision-xt |
| Software, algorithm | pClamp and Clampfit 10.3 | Molecular Devices | RRID:SCR_011323 | https://www.moleculardevices.com/ |
| Software, algorithm | GraphPad Prism 6 | GraphPad Software | RRID:SCR_002798 | https://www.graphpad.com/ |
| Software, algorithm | MiniAnalysis v6.0.9 | Synaptosoft | RRID:SCR_002184 | http://www.synaptosoft.com/MiniAnalysis/ |
| Software, algorithm | Python | Jupyter notebook | RRID:SCR_008394 | https://jupyter.org/ |
| Software, algorithm | ImageJ (Fiji) | National Institutes of Health (NIH) | RRID:SCR_002285 | https://imagej.net/Fiji/Downloads |
| Software, algorithm | Coreldraw X8 | CorelDRAW | RRID:SCR_014235 | https://www.coreldraw.com/en/?trial-delay=none |
| Software, algorithm | Inscopix data processing v1.6.0 | Inscopix | N/A | https://www.inscopix.com/software-analysis |

*Continued*

| Reagent type (species) or resource | Designation | Source or reference | Identifiers | Additional information |
|---|---|---|---|---|
| Other | Microsyringe 10 μL | World Precision Instruments | NANOFIL | https://www.wpiinc.com/ |
| Other | 34 G beveled NanoFil needle | World Precision Instruments | NF34BV-2 | https://www.wpiinc.com/ |
| Other | Animal temperature controller | Physitemp Instruments | TCAT-2LV | https://physitemp.com/ |
| Other | Microsyringe pump controller | KD Scientific | KDS310 | https://www.kdscientific.com/ |
| Other | Cryostat | Leica | CM1900; RRID:SCR_020218 | https://www.leicabiosystems.com/zh/ |
| Other | Microslicer | Dosaka | DTK-1000 | http://www.dosaka-em.jp/products/280/ |
| Other | Axopatch 200B patch clamp amplifier | Molecular Devices | Axopatch 200B; RRID:SCR_018866 | https://www.moleculardevices.com/ |
| Other | Digitizer | Molecular Devices | Digidata 1440A; RRID:SCR_021038 | https://www.moleculardevices.com/ |
| Other | Borosilicate glass with filament | Harvard Apparatus | GC150F-7.5 | https://www.warneronline.com/clark-borosilicate-standard-wall-with-filament |
| Other | Cannula | RWD Life Science | Guide cannula: 0.41 mm in diameter and 5 mm long; Dummy cannula: 0.2 mm in diameter and 5.5 mm long | https://www.rwdstco.com/product-item/single-cannula/ |
| Other | ProView integrated lens | Inscopix | Diameter, 0.5 mm; length, 6.1 mm | https://www.inscopix.com/lenses-viruses |
| Other | Miniscope system | Inscopix | nVista 3.0; RRID:SCR_017407 | https://www.inscopix.com/nvista |

## Animals

Four transgenic mouse lines were used in this study: SST-Cre (stock number [no.] 013044), SST-Cre;Ai14 (SST-Cre line crossed with Ai14 line), PKCδ-Cre (stock no. 011559), and PKCδ-Cre;Ai14 (PKCδ-Cre line crossed with Ai14 line). The SST-Cre line and Ai14 tdTomato reporter (stock no. 007914) were purchased from the Jackson Laboratory. PKCδ-Cre and C57BL/6J mice were purchased from the Mutant Mouse Resource and Research Centers (MMRRC) and National Laboratory Animal Center, respectively. Mice aged 2–5 months of either sex were used in the electrophysiological and behavioral studies. All the mice were bred against the C57BL/6J genetic background. Mice were housed in a 12 hr light-dark cycle and provided food and water ad libitum. Two-month-old mice were injected with a virus and implanted with optical fibers in the CeA. The animals were handled in accordance with national and institutional guidelines. All behavioral procedures were conducted in accordance with the protocol (no. 1080317) approved by the Institutional Animal Care and Use Committee of the National Yang Ming Chiao Tung University.

## Viruses

To specifically express designer receptors exclusively activated by designer drugs (DREADDs) onto the CeA-SST and CeA-PKCδ neurons, we used a recombinant AAV5 carrying hM3Dq or hM4Di conjugated to mCherry in a double-floxed inverted open reading frame (DIO), driven by the human Synapsin I (hSyn) promoter (AAV5-hSyn-DIO-hM3Dq-mCherry or AAV5-hSyn-DIO-hM4Di-mCherry). In the optogenetic experiments, AAV5 carrying channelrhodopsin-2 (ChR2) conjugated to eYFP driven by the CaMKIIα promoter (AAV5-CaMKIIα-ChR2-eYFP) was used to selectively express ChR2 onto the PBN neurons. To silence the neurons, AAV5-hSyn-DIO-hM4Di-mCherry was injected into the CeA region. To activate the neurons, the AAV5-hSyn-DIO-hM3Dq-mCherry virus was injected into the CeA region and the AAV5-CaMKIIα-ChR2-eYFP virus was injected into the PBN region. In addition, a viral vector carrying the red fluorescent protein (AAV5-hSyn-DIO-mCherry) was used as the control. In the

in vivo calcium imaging experiment, the AAV5-Syn-Flex-GCaMP6s-WPRE-SV40 was injected into the CeA region. All viral vectors were purchased from the Vector Core at the University of North Carolina (Chapel Hill, NC, USA) or Addgene Vector Core (Watertown, MA, USA).

## Acid-induced MP model

The model of acid-induced MP (*Sluka et al., 2001*) was used as a preclinical fibromyalgia-like MP model. All mice were briefly anesthetized with isoflurane (4% induction, 1.5–2% maintenance in $O_2$; Halocarbon Laboratories, North Augusta, SC, USA). After anesthesia, the MP and Ctrl mice received 20 µL injections of acidic saline (pH 4.0) and neutral saline (pH 7.2), respectively, on day 0 in the left gastrocnemius muscle. After 3 days (day 3), the same gastrocnemius muscle was re-injected with acidic or neutral saline. The pH value of the 2-(*N*-morpholino) ethanesulfonic acid (MES)-buffered saline (154 mM of NaCl, 10 mM of MES) was used to construct both the acidic and neutral saline, while the pH values were adjusted with 0.1 M of HCl or NaOH.

## Immunohistochemistry

Mice were deeply anesthetized and PBS (0.9% NaCl in 0.01 M of phosphate buffer, pH 7.4) was sequentially perfused through the left ventricle, followed by 30 mL of ice-cold 4% paraformaldehyde in 0.1 M of PBS. The brain was rapidly removed and fixed in 4% paraformaldehyde in 0.1 M of PBS for 6 hr at 4°C, after which it was cryoprotected with 15% sucrose in 0.1 M of PBS for 24 hr at 4°C, followed by 30% sucrose in 0.1 M of PBS for 24 hr at 4°C. Coronal brain sections (45 µm thickness) containing the amygdala and surrounding regions were cut using a cryostat microtome (CM1900, Leica Microsystems, Nussloch, Germany). Sections were treated with 3% $H_2O_2$ for 10 min and then blocked with 0.1% Triton X-100 in TBS containing 2% bovine serum albumin and 2% normal goat serum (Vector Laboratories, Burlingame, CA, USA) for 2 hr at room temperature. To confirm the expression of the virus in the CeA, the slices were stained with a primary antibody: rabbit anti-red fluorescent protein (1:300; Rockland, Limerick, PA, USA) or rabbit anti-PKCδ (1:500; Abcam, Cambridge, UK) or rat anti-SST (1:500; Abcam, Cambridge, UK). After 2 days, the slices were stained with a secondary antibody: goat anti-rabbit Alexa Fluor 594 (1:500; Thermo Fisher Scientific, Waltham, MA, USA) or goat anti-rabbit Alexa Fluor 488 (1:500; Thermo Fisher Scientific, Waltham, MA, USA) or goat anti-rat Alexa Fluor 488 (1:500; Thermo Fisher Scientific, Waltham, MA, USA). The results were examined and imaged under a fluorescence microscope (BX63, Olympus, Tokyo, Japan) or a confocal laser excitation microscope (Leica SP5, Leica Microsystems, Wetzlar, Germany).

## Stereotaxic surgery

Mice were deeply anesthetized with isoflurane (4% induction, 1.5–2% maintenance in $O_2$; Halocarbon Laboratories, North Augusta, SC, USA) and placed in a stereotaxic injection frame (IVM-3000, Scientifica, Uckfield, UK). The injections were performed using the following stereotaxic coordinates: for the CeA, 1.31 mm posterior from bregma, 2.87 mm lateral from the midline on both sides, and 4.72 mm ventral from the cortical surface; for the PBN, 5.1 mm posterior from bregma, 1.2 mm lateral from the midline on both sides, and 3.2 mm ventral from the cortical surface. During all the surgical procedures, the mice were kept on a heating pad (TCAT-2LV CONTROLLER, Physitemp Instruments, Clifton, NJ, USA or Physiological Biological Temperature Controller TMP-5b, Supertech Instruments, Budapest, Hungary) to maintain their surface body temperatures at 34°C. After securing the head with ear bars, 75% ethanol was used to sterilize the surgical area, and the eyes were protected using an ophthalmic gel. For viral injection, we injected 0.35 and 0.5 µL of the viral solution bilaterally into the CeA and PBN, respectively, using a 10 µL NanoFil syringe and a 34-gauge (G) beveled metal needle (World Precision Instruments, Sarasota, FL, USA). The flow rate (0.1 µL/min) was controlled with a nanopump controller (KD Scientific, Holliston, MA, USA). After viral injection, the needle was raised 0.1 mm above the injection site for an additional 10 min to allow the virus to diffuse before being withdrawn slowly. To reach optimal viral expression, all the animals were allowed to recover for at least 4 weeks before conducting behavioral and electrophysiological experiments.

## Cannula implantation for intra-CeA drug application

The cannula for implantation consisted of the guide (27 G, 0.41 mm in diameter and 5 mm long; RWD Life Science, Shenzhen, China) and dummy (0.2 mm in diameter and 5.5 mm long; RWD Life Science,

Shenzhen, China) cannulas. The guide cannulas were placed in the CeA bilaterally (±2.87 mm laterally, 1.31 mm posteriorly, 4.65 mm ventrally). To fix the guide cannulas onto the skull, dental resin cement C&B Super-Bond (Sun Medical, Moriyama, Japan) was applied to the surface of the skull around the cannula for approximately 10 min. After the resin cement hardened, the cannula was removed from the homemade holder and the mice were placed back into their home cages for recovery. One week after recovery, the mice underwent behavioral tests. During the test, the dummy cannulas were replaced by internal cannulas (0.21 mm in diameter and 5.5 mm long; RWD Life Science, Shenzhen, China). We bilaterally injected 0.15 µL of ACSF at a rate of 75 nL/min through the cannulas on days 15 and 18. For the delivery of PGB (Pfizer, New York, NY, USA), we bilaterally injected 0.15 µL of 1 mM PGB at a rate of 75 nL/min through the cannulas on days 16 and 19. For the priming experiment, we bilaterally injected 0.15 µL of ACSF or PGB through the cannulas right after the second acidic saline injection on day 3.

## In vivo calcium imaging

After injection of the virus, the GRIN lens (diameter, 0.5 mm; length, 6.1 mm; Inscopix, Palo Alto, CA, USA) was placed in the CeA (2.87 mm laterally, 1.31 mm posteriorly, 4.72 mm ventrally) region. To fix the lens onto the skull, dental resin cement C&B Super-Bond (Sun Medical, Moriyama, Japan) was applied to the surface of the skull around the lens for approximately 10 min. After the resin cement hardened, the lens was removed from the holder and the mice were placed back into their home cages for recovery. Three to four weeks after injection of the virus, the mice underwent MP induction. Two weeks after MP induction, the calcium activities of the CeA-SST neurons were recorded. Calcium signals were detected using the miniscope system (nVista 3.0, Inscopix). The blue light-emitting diode (LED) power was approximately 1.4 mW at the focal plane. The images (1280×800 pixels) were acquired at a 15 or 20 Hz frame rate. After adapting the miniscope, the calcium imaging was recorded from freely moving mice in the home cage. The analysis was performed using the Inscopix data processing software (version 1.6.0, Inscopix). The constrained nonnegative matrix factorization was applied to extract calcium event traces (ΔF/F) of all the recorded CeA-SST neurons.

## Calcium signal analysis

Before and after the PGB treatment, the frequencies and mean amplitudes of the events were detected using the built-in template searching function (Clampfit 10.3, Molecular Devices, Sunnyvale, CA, USA). For the AUC analysis, in order to eliminate calcium-independent residual noise and to improve the signal-to-noise ratio, the Gaussian kernel filter ($\widetilde{x_t} = \frac{\sum_{i=1}^{n} K(t,i) x_i}{\sum_{j=1}^{n} K(t,j)}$, $K(t,i) = e^{\left(-\frac{(t-i)^2}{2b^2}\right)}$) with b (bandwidth) set as 10 frames was applied to the whole calcium trace of single neurons. Subsequently, calcium signals (ΔF/F) before and after the PGB treatment were concatenated and normalized to the range of 0–1 ($\widetilde{S_t} = \frac{S_t - S_{min}}{(S_{max} - S_{min})}$). For both the BL and PGB trials, calcium traces recorded in the initial 6 min were extracted and resampled to 10 Hz by interpolation. Spontaneous activity of a neuron was measured via calibrating its AUC per second from the normalized calcium activity of a given trial. The standard deviation of all neuron changes in the AUC per second following PGB application ($\sigma_{dev}$) was arbitrarily used as a cutoff threshold to identify PGB-responsive neurons. After application of PGB, those with ΔAUC greater than $1 \times \sigma_{dev}$ were arbitrarily defined as either the PGB excitation or inhibition group. The remaining neurons were categorized as the no effect group.

## Slice preparation and electrophysiology

After the behavioral tests, AAV-injected mice were sacrificed and acute coronal brain slices of 300 µm thickness were cut using a vibratome (DTK-1000, Dosaka, Kyoto, Japan) in ice-cold sucrose saline containing the following (in mM): 87 NaCl, 25 NaHCO₃, 1.25 NaH₂PO₄, 2.5 KCl, 10 glucose, 75 sucrose, 0.5 CaCl₂, and 7 MgCl₂. Slices were allowed to recover in an oxygenated (95% O₂ and 5% CO₂) sucrose saline-containing chamber at 34°C for 30 min before being maintained at room temperature until recording. During the experiment, slices were transferred to a submerged chamber and perfused with oxygenated ACSF containing the following (in mM): 125 NaCl, 25 NaHCO₃, 1.25 NaH₂PO₄, 2.5 KCl, 25 glucose, 2 CaCl₂, and 1 MgCl₂. The expression of the virus or tdTomato expression was confirmed by red or green fluorescence and the neurons in the CeA were visually selected for recordings under an infrared differential interference contrast microscope (BX51WI, Olympus,

Tokyo, Japan) equipped with an LED source (505 and 590 nm, LED4D162, controlled by DC4104 driver, Thorlabs, Newton, NJ, USA). For optical stimulation, ChR2-expressing neurons were excited by 470 nm LED light (driven by DC4104 driver, Thorlabs, Newton, NJ, USA). Whole-cell patch-clamp recordings were conducted using an Axopatch 200B amplifier (Molecular Devices, Sunnyvale, CA, USA). Recording electrodes (3–6 MΩ) were pulled from borosilicate glass capillaries (outer diameter, 1.5 mm; 0.32 mm wall thickness; Harvard Apparatus). The glass microelectrodes were filled with a low Cl⁻ internal solution containing the following (in mM): 136.8 K-gluconate, 7.2 KCl, 0.2 EGTA, 4 $MgATP$, 10 HEPES, 7 $NA_2$-phosphocreatine, 0.5 $Na_3GTP$, and 0.4% biocytin (wt/vol; ~310 mOsm/L). The pipette capacitance was compensated. Signals were low-pass filtered at 5 kHz and sampled at 10 kHz using a digitizer (Digidata 1440 A; Molecular Devices, Sunnyvale, CA, USA). In the ex vivo slice recordings, the following antagonists were added to the ACSF: SR-95531 (1 µM; Abcam), CGP-55845 (1 µM; Tocris), and kynurenic acid (2 mM; MERCK) to block the $GABA_A$, $GABA_B$, AMPA/NMDA receptors, respectively. Tetrodotoxin (TTX; 1 µM; Tocris) was applied to block sodium channels in the experiments of the mEPSC recording. CNO (5 µM; MERCK) was used to activate the DREADDs. PGB (500 µM; Pfizer, New York, NY, USA) was used to examine its potential effects on synaptic transmission at the PBN to the CeA-SST or CeA-PKCδ neuron synapses.

## Behavioral tests

Mice were handled for at least 3 days before the behavioral tests (*Hurst and West, 2010*). All the behavioral tests were conducted during the light period of the light-dark cycle. Mice were moved to the behavioral room with dim light at least 30 min before performing the experiments. In chemogenetic experiments, CNO was freshly dissolved in injection saline (10% v/v DMSO in 0.9% NaCl) and i.p. injected at 5 mg/kg of body weight. Behavioral tests were performed approximately 50 min following CNO injection.

### Von Frey filament test

Mechanical hypersensitivity was assessed using the von Frey filament test. A series of von Frey filaments of increasing stiffness (0.04–1.4 g) were applied to the plantar surface of both the hindpaws. Each filament was applied five times and the lowest force that caused at least three withdrawals out of the five stimuli was considered the threshold (g) (*Blackburn-Munro and Jensen, 2003*; *Hao et al., 1999*).

### Marble burying test

The clean cage (height, 12.5 cm; length, 28 cm; width, 17 cm) was filled approximately 6 cm high with bedding. Twenty-four glass marbles (approximately 1.5 cm in diameter) were evenly spaced on the top of the bedding with an approximately 3 cm distance between each pair of marbles. Mice were placed individually into the cages and left undisturbed for 30 min (*Chang et al., 2017*). Marbles were considered buried if at least two-thirds of their surface was covered by bedding.

### L/D box test

The test apparatus consisted of a two-compartment light-dark box (height, 40 cm; length, 42 cm; width, 42 cm) connected by a central opening at the floor level. Mice were placed individually in the center of the brightly lit side of the box and left undisturbed for 10 min of exploration. Transitions between the two compartments and total locomotor activity were recorded using the Tru-scan 2.0 system (Coulbourn Instrument, Allentown, PA, USA).

### EPM

The EPM is a common anxiety test that produces approach avoidance conflict (*Walf and Frye, 2007*). The EPM apparatus consisted of two open arms (length, 30 cm; width, 5 cm) and two closed arms (length, 30 cm; width, 5 cm; height, 25 cm) extending from the intersection zone (5 cm × 5 cm). The EPM was elevated 50 cm from the floor. The recording camera was placed above the maze. Mice were placed in the center of the intersection zone and then allowed to freely explore for 10 min. The open-arm time and total travel distance were measured using the video tracking software EthoVision XT 13 (Noldus Information Technology, Leesburg, VA, USA).

## Sociability test

The three-chamber sociability test has been successfully employed to study social affiliation in several mouse lines (*Yang et al., 2011*). The social approach apparatus was an open-topped plastic box (height, 22 cm; length, 52.5 cm; width, 42.5 cm) divided into three chambers by two clear walls. The center compartment (length, 16.5 cm) was smaller than the other two compartments, which were equal in size to each other (length, 18 cm). The dividing walls had retractable doorways, allowing access into each chamber. A wire cup (bottom diameter, 5 cm) was used to confine the novel mice. Mice were housed alone for 24 hr before the test and underwent the test in a darkened room. The lighting in the two side chambers was maintained at approximately 5–6 lux calibrated by a hand-held lux meter. Mice were habituated to the inverted wire cup for two 15 min sessions before the test session. Test mice were confined in the center chamber at the beginning of each phase for 10 min for habituation. During the habituation phase, each of the two side chambers contained an inverted empty wire cup. To initiate each 10 min phase, the doorways to the side chambers were opened, and the mice were allowed to explore freely. During the sociability phase, an unfamiliar mouse was enclosed in one of the wire cups in the side chambers. The time spent in each chamber and time spent exploring the enclosed novel mice or empty cups were recorded with a camera mounted overhead and analyzed using the EthoVision XT software 13 (Noldus Information Technology, Leesburg, VA, USA).

## FST

The FST consisted of a transparent acrylic cylindrical container (height, 25 cm; width, 10 cm) filled with water to a height of 16 cm (*Can et al., 2012*; *Castagné et al., 2010*; *Slattery and Cryan, 2012*). The mouse was placed in the water maintained at 20–22°C for 6 min. We measured the immobility time that was indicative of depression using the video tracking software EthoVision XT 13 (Noldus Information Technology, Leesburg, VA, USA).

## Data analysis and statistics

Data were analyzed using Clampfit 10.3 (Molecular Devices, Sunnyvale, CA, USA), MiniAnalysis (Synaptosoft, Inc, Fort Lee, NJ, USA), GraphPad Prism 6 (GraphPad Software, Inc, San Diego, CA, USA) and a custom-made program written in Python. To define the effect of PGB, we classified PGB-inhibited neurons as having the EPSC inhibition $>\times 1$ $SD_{BL}$. Statistical significance was tested using the two-way analysis of variance (ANOVA) with the Tukey's post hoc test, two-way ANOVA with the Bonferroni's multiple comparison test, Kolmogorov–Smirnov test, Wilcoxon matched-pairs signed rank test, or Mann–Whitney test at the significance level (p) indicated. Data are presented as mean ± standard error of mean (SEM). Significance levels were set at $p < 0.05$ (*), $p < 0.01$ (**), $p < 0.001$ (***), or $p < 0.0001$ (****).

## Acknowledgements

We thank Dr SC Lin, Dr PH Chiang (National Yang Ming Chiao Tung University, Taiwan), Dr A Dominique (University of Liege, Belgium), and Dr P Turko (Charite University Medicine Berlin, Germany) for commenting on earlier versions of the manuscript and all the members of the Lien lab for their insightful discussions. We also thank Pfizer for the supply of pregabalin. Funding: This work was financially supported by the Brain Research Center, National Yang Ming Chiao Tung University from the Featured Areas Research Center Program within the framework of the Higher Education Sprout Project by the Ministry of Education in Taiwan, National Health Research Institutes (NHRI-EX111-11135NI), and Ministry of Science and Technology (MOST 108–2320-B-010-026-MY3, MOST 109–2926-I-010–506, MOST 110–2321-B-010-006, MOST 111-2321-B-A49-005) in Taiwan.

## Additional information

### Funding

| Funder | Grant reference number | Author |
|---|---|---|
| National Health Research Institutes | NHRI-EX111-11135NI | Cheng-Chang Lien |
| Ministry of Science and Technology, Taiwan | MOST 108-2320-B-010-026-MY3 | Cheng-Chang Lien |
| Ministry of Science and Technology, Taiwan | MOST 109-2926-I-010-506 | Cheng-Chang Lien |
| Ministry of Science and Technology, Taiwan | MOST 110-2321-B-010-006 | Cheng-Chang Lien |
| Ministry of Science and Technology, Taiwan | MOST 111-2321-B-A49-005 | Cheng-Chang Lien |

The funders had no role in study design, data collection and interpretation, or the decision to submit the work for publication.

### Author contributions

Yu-Ling Lin, Conceptualization, Formal analysis, Validation, Investigation, Visualization, Writing - original draft; Zhu-Sen Yang, Wai-Yi Wong, Formal analysis, Validation, Investigation; Shih-Che Lin, Formal analysis; Shuu-Jiun Wang, Shih-Pin Chen, Jen-Kun Cheng, Hui Lu, Conceptualization; Cheng-Chang Lien, Conceptualization, Supervision, Funding acquisition, Writing - review and editing

### Author ORCIDs

Jen-Kun Cheng ⓘ http://orcid.org/0000-0003-2384-7856
Cheng-Chang Lien ⓘ http://orcid.org/0000-0002-6692-9942

### Ethics

The animals were handled in accordance with national and institutional guidelines. All behavioral procedures were conducted in accordance with the protocol (No. 1080317) approved by the Institutional Animal Care and Use Committee (IACUC) of the National Yang Ming Chiao Tung University.

### Decision letter and Author response

Decision letter https://doi.org/10.7554/eLife.78610.sa1
Author response https://doi.org/10.7554/eLife.78610.sa2

## Additional files

### Supplementary files

• MDAR checklist

• Transparent reporting form

### Data availability

Source data for all figures have been provided.

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
