## [Editor Report]

This study investigates the role of the central amygdala (CeA) on the mechanisms underlying chronic pain. An acid-induced muscle pain (AIMP) mouse model is used. Changes of excitatory synaptic and intrinsic excitability of GABAergic somatostatin-expressing (SST+) neurons of the CeA are detected in the chronic MP model. The manuscript is extensive, well-illustrated and contains valuable findings that are likely to inspire future work to progress the understanding of the GABAergic system on patho-physiological events.

---

## [Decision Letter]

**Decision letter after peer review:**

Thank you for submitting your article "Cellular Mechanisms Underlying Central Sensitization in a Mouse Model of Chronic Muscle Pain" for consideration by *eLife*. Your article has been reviewed by 3 peer reviewers, including Marco Capogna as Reviewing Editor and Reviewer #1, and the evaluation has been overseen by Gary Westbrook as the Senior Editor.

Essential revisions:

The reviewers and editors agreed that this manuscript is interesting, but they also raised criticism that needs to be taken into consideration. We would like to ask you to prepare a revised version of the manuscript and re-submit. The ideal deadline for re-submission for *eLife* is two months, but some more experiments will be required. Thus the authors should take the time required to address the issues below in a convincing manner as well as respond to the comments of the original reviews.

1) The investigation of the direct monosynaptic excitatory input from the PBN to CeA neurons should be completed by recordings from PKCd+ neurons. Furthermore, more statistical analyses of the data already submitted should be performed, as explained by Reviewer 1.

2) Increase the number of animals used with chronic muscle pain treated with PGB for the EPM and sociability behavioral tests.

3) Provide control data on post-hoc images of the virus injection/expression and co-localization with cell type specific markers; refer to previously published data or publicly available data (Allen Inst) on a2d VGCC subunit expression in parabrachial -CeA.

4) Better discuss: acute PGB effect on the mechano-sensitivity versus emotional behavior; mechanisms involved in changes in intrinsic excitability of neurons in the MP model; pre- versus pot-synaptic changes detected in the MP model; the source(s) of the spontaneous synaptic events.*Reviewer #1 (Recommendations for the authors):*

1) The direct monosynaptic excitatory input from the parabrachial nucleus (PBN), which is enriched with the calcium channel α2δ subunit, onto CeA neurons, is directly investigated by presynaptic optogenetic stimulation and postsynaptic recordings of SST neurons (Figure 5B). I think this is an important approach that should be further exploited. From the data submitted, it seems that such evoked EPSCs are about 3 times bigger in MP mice compared to controls. Provided that optogenetically-evoked EPSCs from control and MP mice could be appropriately compared, e.g. through normalization, is this difference statistically significant?

It would be also useful to statistically compare the inhibition of the evoked EPSCs by PGB in the two genotypes. Finally, could the Authors perform some more experiments and report the results of a similar experiment by recording from PKCδ neurons, and statistically compare it to the evoked EPSCs recorded in SST neurons?

2) What mechanisms the Authors envisage to account for by the changes in intrinsic excitability in SST+ and PKCδ neurons observed in MP mice? This aspect should be discussed.

3) The Authors reported effects on amplitude and/or frequency of sEPSCs and mEPSCs onto CeA recorded neurons seem to suggest a presynaptic mechanism of action. In contrast, the effect of PGB on calcium transient revealed by 1P deep calcium imaging may be interpreted as pre and/or postsynaptic.

The Authors should make a more definitive conclusion whether they think their data suggest a presynaptic only or also a postsynaptic mechanism of action. It may be difficult to draw any conclusion from sEPSCs since these events are probably due to the release of multiple synaptic vesicles (e.g., mean amplitude of sEPSCs is bigger than the mean amplitude of mEPSCs).

4) The Authors should discuss the possible source(s) of the spontaneous synaptic events. Do they only originate from PBN terminals or also from BLA principal neurons or elsewhere? Is an analysis of mEPSCs kinetics compatible with an homogeneous pool of presynaptic terminals as source of the m/sEPSCs or rather SST+ and PKCδ+ neurons are biophysically too compact to make out the presence of heterogeneous excitatory inputs perhaps impinging on different spatial domains of the neurons' dendritic trees?

5) The paw withdraw effect could be replicated by either reducing the CeA SST+ cell activity or by elevating PKCd+ cell activity, but this is not the case for the anxiety-like affective behaviors (Figure 4, Figure 4 S2). Therefore, the PGB effect on the increased paw withdraw threshold and emotion behaviors in the MP mouse model may be caused by distinct circuit mechanisms occurring simultaneously.

To further elucidate their observation, the Author should test the acute PGB effect on the mechano-sensitivity and emotional behavior separately using a wild type mice or should at least acknowledge this possibility in the discussion.

*Reviewer #2 (Recommendations for the authors):*

My specific issues are outlined below:

1) Why does the number of animals tested in the different behavioural paradigms (relative to Figures 2 and 4) vary from test to test? Were these different cohorts for each behavioural test?

2) The size of the cohorts in which PGB was tested (and the relative controls) after induction of chronic muscle pain (Figure 2A-E) is suboptimal for the EPM and sociability tests, also in view of the known relatively high interindividual variability, and should, therefore, be increased.

3) It should be made more clear in the text that in vivo ca^2+^ imaging experiments were performed in mice with chronic muscle pain. Why did the authors did not test the effect of PGB on SST+ neuron activity in control animals? This would have strengthened their findings. Moreover, they could have better exploited their study by testing the effect of PGB on SST+ neuron activity during a von Frey filament stimulation, as this test offers good behavioral bouts in which to analyse ca^2+^ transients.

*Reviewer #3 (Recommendations for the authors):*

I have the following comments, which require more experiments to address in order to make the manuscript more convincing.

1, Figure 3 shows recording results from CeA-SST and CeA-PKCd neurons in the MP mice. Since these neurons at baseline consist of different firing phenotypes (spontaneous, late-firing, and regular firing), it would be useful to know whether all firing types show the same changes in synaptic transmission and excitability in the MP model.

Notably, in Wilson et al., 2019 paper, they found that CeA neurons require at least 50-80 pA current injection to induce the spikes, whereas in this manuscript, CeA neurons require a much lower current injection to fire. Could the author provide some explanation to this difference?

2, The number of animals in each test is quite different. For instance, in Figure 4E, n=10 for the mCherry group and n=52 for the hM4Di group; this is a big difference to compare 2 groups. Also, in Figure 4 supplement 2, in the CeA-PKCd experiment, n=4 for the mCherry group and n=13 for the hM3q group. These numbers are significantly fewer than those for CeA-SST hM4Di group (n=52). Please explain how the numbers of animals were assigned to each test? Did the author exclude some animals from some tests?

3, The authors did not show any post-hoc images of the virus injection/expression and co-localization with cell type specific markers for all the functional manipulation and imaging studies. These should be provided to substantiate the results.

4, In Figure 4 supplement 1B: Chemogenetic inhibition of CeA-SST neurons does not affect the paw withdrawal response in the naïve mice. In contrast, the authors themselves showed in their recent paper that optogenetic inhibition of CeA-SST neurons increased paw withdrawal response (Chen et al., 2022, Pain). Please reconcile these two different results in the same lab.

5, There is no data on a2d VGCC subunit expression in parabrachial or CeA in the chronic MP model in this manuscript. Given so much emphasis on PGB results, this information is crucial and is missing here.

6, The authors performed the in vivo calcium imaging of CeA-SST neurons in response to PGB. It would be more informative to know how CeA-SST neurons or CeA-PKCd neurons respond to 1st and 2nd acid injection.

[Editors’ note: further revisions were suggested prior to acceptance, as described below.]

Thank you for resubmitting your work entitled "Cellular Mechanisms Underlying Central Sensitization in a Mouse Model of Chronic Muscle Pain" for further consideration by *eLife*. Your revised article has been evaluated by Gary Westbrook (Senior Editor) and a Reviewing Editor.

The manuscript has been improved but there are some remaining issues that need to be addressed, as outlined here and in the novel comments by the reviewers:

Essential revisions:

All the reviewers acknowledged that the Authors replied satisfactorily to most of their comments and that the first revised version is improved. However, they also concur that the manuscript still needs some significant improvements.

Specifically:

1) The text is often hard to read and should be carefully checked to improve readability. If needed, a professional writer could be helpful.

2) All the reviewers acknowledged the additional excel database and your explanations regarding the data in Figure 3 – supplement 2 B-D. These data were submitted to support the electrophysiological results – that SST+ neuron excitability is selectively enhanced in MP mice. (See also comments of Reviewer 3 in their original review)

However, some improvements are needed:

a) We think that you pooled the data from "all slices" together at a given AP position relative to Bregma, to generate the plots and calculate SEM. However, we think you should re-calculate the mean at each AP position for each animal first, then average across animals and calculate SEM. In this way, the data would be averaged within an animal and the variance would be calculated between animals in order to assess the robustness of the results;

b) Explain why the number of slices at a given AP position varies dramatically across animals (some animals have 1 slice, some have 5-9 slices at a given AP-position). Furthermore, how it could be that in some mice the AP position has 5-9 slices, whereas from the method one can calculate that a maximum of 3 slices can be obtained for a given AP position;

c) The number of samples for each bregma level and animal should be similar and more control data should be collected and included since currently they show large variability;

d) Statistical analysis should be improved, e.g. a two-factor analysis of variance (with equal replications) would be a good way to analyze these data.

*Reviewer #1 (Recommendations for the authors):*

I think that the Authors have done a good job in revising the manuscript according to my suggestions.

Specifically:

1) They further investigated and analyzed synaptic transmission at the PBN input to the CeA. The data suggest that the PBN input is not the major potentiated input onto the CeA-SST neurons in the MP mice. They also compared the effect of PGB inhibition between the Ctrl and MP groups and found it similar. Then, they tested the effect of PGB on synaptic transmission at the PBN-CeAPKCδ neuron synapses. Approximately 30% CeA-PKCδ neurons were sensitive to PGB inhibition in the Ctrl mice and approximately 63% CeA-PKCδ neurons were sensitive to PGB inhibition in the MP mice.

2) Potential changes in intrinsic excitability of CeA-SST neurons are discussed in the revised version of the manuscript.

3) The Authors strengthen the interpretation of enhancement of vesicle release at the inputs onto the CeA-SST neuron synapses along with the increased excitability to account for the central sensitization of the CeA-SST neurons in MP.

4) They addressed the issue of PGB mediated increase in paw withdraw threshold and emotion behaviors.

*Reviewer #2 (Recommendations for the authors):*

The work by Lin and coworkers addresses the important and of broad interest process that underlies the progression of transient pain into persistent pain. This study used a number of sophisticated techniques to dissect the synaptic and cellular mechanisms that occur in the mouse central amygdala (CeA) during the development of chronic muscle pain. In particular, the study reveals shifts in opposite directions of the activity of somatostatin-expressing (SST+) and PKCδ+ neurons. Pharmacological and chemogenetic manipulation of these neurons prevented the chronification of pain and of the comorbid affective behaviors, although with important differences between these two neuronal classes.

In their revised manuscript, the authors have made a number of significant changes by which they could adequately address my main concerns. In particular, the additional experimental data aimed at increasing the sample size of the behavioural experiments has appreciably improved the manuscript.

While the text has improved, it still contains several syntax and grammar errors that need to be corrected.

*Reviewer #3 (Recommendations for the authors):*

In the revised manuscript, the authors addressed most of my previous concerns. In their response to my question of conflicting results for inhibition of CeA-SST neurons from their prior publication, they responded that "It seems that different illumination protocols resulted in opposite behaviors". "5 min optogenetic inhibition of CeA-SST reduced mechanical sensitivity, while 10 min illumination increased mechanical sensitivity". I honestly cannot think of any known mechanism that could explain opposing effect induced by 5min vs 10min optogenetic inhibition. I give up.

I do have some questions for Figure 3—figure supplement 2

In D, the number of SST(-)pERK(+) in the Ctrl (-1.82 mm) is above 40, while in B, the number of total pERK(+) in the Ctrl (-1.82mm) is about 30. This is weird. How could total number of pERK^+^ cells in B being less than that in D (a subset of neurons)?

Authors concluded in Line 212-213, "increased pERK-positive cells are mostly CeA-SST cells." From the results shown in C and D, it's difficult to be convinced of this statement: the number of SST(+) pERK(+) cells increased from 1.1 to 3.7, while SST(-) pERK(+) cells increased from 18 to 28, even though statistics appeared to be not significant. I believe that instead of comparing the averaged number across sections, comparing the total number of cells in the CeA makes more sense.

The "gray" bars for "control" samples in panel B (total pERK^+^) and in panel D (SST-/pERK^+^) appear identical. Could there be an image duplication here?

---

## [Author Response]

Essential revisions:1) The investigation of the direct monosynaptic excitatory input from the PBN to CeA neurons should be completed by recordings from PKCd+ neurons. Furthermore, more statistical analyses of the data already submitted should be performed, as explained by Reviewer 1.

In the revised version of the manuscript, we completed the investigation of the direct monosynaptic excitatory input from the PBN to the CeA-PKCδ neurons (Figure 5 —figure supplement 1). As the reviewer #1 suggested, we also performed additional statistical analyses of the data (see our response to the reviewer #1).

2) Increase the number of animals used with chronic muscle pain treated with PGB for the EPM and sociability behavioral tests.

In the revised version of the manuscript, we increased the number of experiments in Figure 2B (Ctrl-ACSF, from n = 5 to 6; Ctrl-PGB, from n = 4 to 6; MP-ACSF, from n = 5 to 7; MP-PGB, from n = 6 to 7) and 2E (Ctrl-ACSF, from n = 5 to 6; Ctrl-PGB, from n = 5 to 7; MP-ACSF, from n = 4 to 6; MP-PGB, from n = 5 to 6).

3) Provide control data on post-hoc images of the virus injection/expression and co-localization with cell type specific markers; refer to previously published data or publicly available data (Allen Inst) on a2d VGCC subunit expression in parabrachial -CeA.

We added images of virus expression (AAV5-hSyn-hM4Di-mCherry or AAV5-hSyn-hM3Dq-mCherry) colocalized with cell type specific markers (SST or PKCδ) in Figure 4B and Figure 4 —figure supplement 4B in the revised submission. The colocalization of specific markers with virus expression was summarized in Figure 4 —figure supplement 1 and 4B in the revised submission.

Regarding the α2δ VGCC subunit expression, we cited a previous study by Cole et al. (2005), which demonstrated the anatomical distribution of α2δ VGCC subunit expression across the whole rat brain (page 5, line 1 – 2). Moreover, we provided the published data by Allen Brain Institute, which demonstrated α2δ1 and α2δ2 VGCC subunit expression in the parabrachial nucleus in our supplement file (Figure 1 —figure supplement 2).

4) Better discuss: acute PGB effect on the mechano-sensitivity versus emotional behavior; mechanisms involved in changes in intrinsic excitability of neurons in the MP model; pre- versus pot-synaptic changes detected in the MP model; the source(s) of the spontaneous synaptic events.

In the revised version of manuscript, we included discussions on acute PGB effect on the mechano-sensitivity versus emotional behavior (Figure 2 —figure supplement 1; page 9, line 16 – 33), mechanisms involved in changes in intrinsic excitability of neurons in the MP model (page 10, line 24 – 37), pre- versus post-synaptic changes detected in the MP model (page 9, line 35 – 49; page 10, line 1 – 23), and the source(s) of the spontaneous synaptic events (page 10, line 11 – 23). Please also refer to our response to the Reviewer #1 (point #1 – 5).

Reviewer #1 (Recommendations for the authors):1) The direct monosynaptic excitatory input from the parabrachial nucleus (PBN), which is enriched with the calcium channel α2δ subunit, onto CeA neurons, is directly investigated by presynaptic optogenetic stimulation and postsynaptic recordings of SST neurons (Figure 5B). I think this is an important approach that should be further exploited. From the data submitted, it seems that such evoked EPSCs are about 3 times bigger in MP mice compared to controls. Provided that optogenetically-evoked EPSCs from control and MP mice could be appropriately compared, e.g. through normalization, is this difference statistically significant?It would be also useful to statistically compare the inhibition of the evoked EPSCs by PGB in the two genotypes. Finally, could the Authors perform some more experiments and report the results of a similar experiment by recording from PKCδ neurons, and statistically compare it to the evoked EPSCs recorded in SST neurons?

In revision, we further investigated and analyzed synaptic transmission at the PBN input to the CeA. First, we found the average EPSC amplitude was 1.5 times greater in the MP mice compared to the Ctrl mice. However, the change did not reach the statistical significance (Ctrl, 73.9 ± 17.9 pA, n = 8; MP, 110.7 ± 20.0 pA, n = 8; Mann–Whitney test, U = 17, n.s., non-significant, *p* = 0.13; Figure 5 —figure supplement 2A). Moreover, the PPR of the light-evoked EPSCs recorded at the PBN-CeA-SST neuron synapses showed no change in the Ctrl and MP groups (Ctrl, 0.65 ± 0.04, n = 8; MP, 0.60 ± 0.02, n = 8; Mann–Whitney test, U = 25, n.s., non-significant, *p* = 0.49). These results may suggest that the PBN input is not the major potentiated input onto the CeA-SST neurons in the MP mice. However, it is worth noting that the CeA-SST neurons were randomly selected for recording. It is very possible that not all recorded CeA-SST neurons were sensitized neurons. Therefore, we need to carefully interpret our data.

Second, we compared the effect of PGB inhibition between the Ctrl and MP groups (Figure 5C). Bath application of PGB (500 µM) decreased the amplitude of light-evoked EPSCs in some CeA-SST neurons of both the Ctrl and MP mice (Figure 5B). Overall, the percentage of PGB-sensitive neurons was only slightly increased in the MP mice (Figure 5C left; Ctrl mice, 37%; 8 cells from 5 mice vs. MP mice, 63%; 8 cells from 4 mice; Chi-square test, *p* = 0.317) and the degree of inhibition between these two groups was similar (Ctrl, 24.9 ± 5.3 %, n = 3; MP, 30.1 ± 5.5 %, n = 5; Mann–Whitney test, U = 5, n.s., non-significant, *p* = 0.5).

Finally, we tested the effect of PGB on synaptic transmission at the PBN-CeA-PKCδ neuron synapses. The amplitude of the light-evoked EPSCs recorded at the PBN-CeA-PKCδ neuron synapses was decreased in the MP group (Ctrl, 134.2 ± 13.2 pA, n = 10; MP, 75.7 ± 14.5 pA, n = 8; Mann–Whitney test, U = 12, **p* = 0.01; Figure 5 —figure supplement 2C). However, the paired-pulse ratio of the light-evoked EPSCs recorded at the PBN-CeA-PKCδ neuron synapses was not different between the Ctrl and MP groups (Ctrl, 0.62 ± 0.06, n = 10; MP, 0.52 ± 0.08, n = 8; Mann–Whitney test, U = 28, n.s., non-significant, *p* = 0.31; Figure 5 —figure supplement 2D). Compared to the amplitude of the light-evoked EPSCs at the PBN-CeA-SST neuron synapses, the light-evoked EPSCs amplitude of the CeA-PKCδ neurons was increased in the Ctrl mice (SST, 73.9 ± 17.9 pA, n = 8; PKCδ, 134.2 ± 13.2 pA, n = 10; Mann–Whitney test, U = 13, **p* = 0.016; Figure 5 —figure supplement 2E), but showed no significant change in the MP mice (SST, 110.7 ± 20.0 pA, n = 8; PKCδ, 75.7 ± 14.5 pA, n = 8; Mann–Whitney test, U = 19, *p* = 0.19; Figure 5 —figure supplement 2F). In addition, the paired-pulse ratio of the light-evoked EPSCs recorded at the synapses of the PBN-CeA-PKCδ neurons and PBN-CeA-SST neurons were not different between the Ctrl (SST, 0.65 ± 0.04, n = 8; PKCδ, 0.62 ± 0.06, n = 10; Mann–Whitney test, U = 18, n.s., non-significant, *p* = 0.054; Figure 5 —figure supplement 2G) and MP groups (SST, 0.60 ± 0.02, n = 8; PKCδ, 0.52 ± 0.08, n = 8; Mann–Whitney test, U = 28, n.s., non-significant, *p* = 0.7; Figure 5 —figure supplement 2H). Bath application of PGB decreased the amplitude of light-evoked EPSCs in the CeA-PKCδ neurons of both the Ctrl and MP mice (Figure 5 —figure supplement 1B, D). Approximately 30% CeA-PKCδ neurons were sensitive to PGB inhibition in the Ctrl mice and approximately 63% CeA-PKCδ neurons were sensitive to PGB inhibition in the MP mice (Figure 5 —figure supplement 1C). Furthermore, the degree of inhibition between these two groups were similar (Figure 5 —figure supplement 1C). Consistent with the effect of PGB on presynaptic α2δ subunit of VGCCs, the reduction of EPSCs was concomitant with an increase in the paired-pulse ratio (Figure 5 —figure supplement 1E).

2) What mechanisms the Authors envisage to account for by the changes in intrinsic excitability in SST+ and PKCδ neurons observed in MP mice? This aspect should be discussed.

Several molecular mechanisms underlying aberrant plasticity of sensitized neurons were reported in different chronic pain animal models (Min et al. 2011; Sun and Chen, 2016; Cheng et al., 2011; Fu et al., 2008). These changes include synaptic and intrinsic plasticity. We reported that increased spike number in response to current injection and reduced rheobase in the CeA-SST neurons in the MP mice (Figure 3), whereas their input resistance and resting potential were unaltered (Figure 3 —figure supplement 1). Therefore, we speculate that some active conductances such as voltage-gated sodium channels are upregulated in the CeA-SST neurons of the MP mice. It is worth noting that increased excitability was specific to early-spiking (ES) neurons, the subtype of CeA-SST neurons in the MP mice. Conversely, the decreased excitability was specific to the ES type of PKCδ neurons. According to previous studies, modifications of intrinsic excitability are likely dependent on the PKC-ERK pathway (Min et al. 2011; Cheng et al., 2011). We discussed the possible mechanisms in Discussion (page 10, line 7 – 37).

References:

Min MY, Yang HW, Yen CT, Chen CC, Chen CC, Cheng SJ. (2011). ERK, synaptic plasticity and acid-induced muscle pain. Commun Integr Biol *4*, 394–396. Doi: 10.4161/cib.4.4.15694.

Sun, WH, and Chen, CC. (2016). Roles of Proton-Sensing Receptors in the Transition from Acute to Chronic Pain. J Dent Res *95*, 135–142. Doi: 10.1177/0022034515618382.

Cheng SJ, Chen CC, Yang HW, Chang YT, Bai SW, Chen CC, Yen CT, Min MY. (2011). Role of extracellular signal-regulated kinase in synaptic transmission and plasticity of a nociceptive input on capsular central amygdaloid neurons in normal and acid-induced muscle pain mice. J Neurosci *31*, 2258–2270. Doi: 10.1523/JNEUROSCI.5564-10.2011.

Fu Y, Han J, Ishola T, Scerbo M, Adwanikar H, Ramsey C, Neugebauer V. (2008). PKA and ERK, but not PKC, in the amygdala contribute to pain-related synaptic plasticity and behavior. Mol Pain *4*, 26. Doi: 10.1186/1744-8069-4-26.

3) The Authors reported effects on amplitude and/or frequency of sEPSCs and mEPSCs onto CeA recorded neurons seem to suggest a presynaptic mechanism of action. In contrast, the effect of PGB on calcium transient revealed by 1P deep calcium imaging may be interpreted as pre and/or postsynaptic.The Authors should make a more definitive conclusion whether they think their data suggest a presynaptic only or also a postsynaptic mechanism of action. It may be difficult to draw any conclusion from sEPSCs since these events are probably due to the release of multiple synaptic vesicles (e.g., mean amplitude of sEPSCs is bigger than the mean amplitude of mEPSCs).

Our data showed that PGB suppressed synaptic transmission and increased the PPR at the PBN-CeA-SST neuron synapses (Figure 5). Therefore, we interpreted the suppression of calcium transients in the CeA-SST neurons by PGB was mediated by presynaptic inhibition (also see page 8, line 5 – 7).

Regarding the mechanism of synaptic plasticity, we reported that both the sEPSC and mEPSC frequency were increased in the CeA-SST neurons. Therefore, the enhancement of vesicle release at the inputs onto the CeA-SST neuron synapses along with the increased excitability might mainly account for the central sensitization of the CeA-SST neurons in MP (page 9, line 35 – 49; page 10, line 1 – 23).

4) The Authors should discuss the possible source(s) of the spontaneous synaptic events. Do they only originate from PBN terminals or also from BLA principal neurons or elsewhere? Is an analysis of mEPSCs kinetics compatible with an homogeneous pool of presynaptic terminals as source of the m/sEPSCs or rather SST+ and PKCδ+ neurons are biophysically too compact to make out the presence of heterogeneous excitatory inputs perhaps impinging on different spatial domains of the neurons' dendritic trees?

In revision, we discussed the possible sources of the spontaneous synaptic events (see below; also page 10, line 11 – 23 in the Discussion). Several excitatory inputs impinge onto CeA neurons (Fadok et al., 2018). The CeA receives nociceptive inputs from the dorsal horn via the PBN and affect-related information from the LA (Li et al., 2020; Wilson et al., 2019; Ikeda et al., 2007). The PBN-CeA synaptic potentiation is consolidated in chronic neuropathic pain (Ikeda et al., 2007). In addition, BLA-CeA synapses are potentiated in chronic neuropathic pain (Ikeda et al., 2007). The PVT, which is a structure that is readily activated by both physical and psychological stressors, also projects to the CeA (Penzo et al., 2015). Taken together, our results revealed that the net synaptic efficacy of the PBN-CeA, PVT-CeA or BLA-CeA synapses onto CeA-SST neurons were strengthened, whereas the synapses onto CeA-PKCδ neurons were weakened in chronic muscle pain.

Our preliminary analysis of sEPSC decay time revealed at least two populations of sEPSCs, suggesting heterogeneous excitatory inputs impinging on different spatial domain of the neuron’s dendrites (Figure 3 —figure supplement 3).

References:

Fadok JP, Markovic M, Tovote P, Lüthi A. (2018). New perspectives on central amygdala function. Curr Opin Neurobiol *49*, 141–147. Doi: 10.1016/j.conb.2018.02.009.

Li JN, and Sheets PL. (2020). Spared nerve injury differentially alters parabrachial monosynaptic excitatory inputs to molecularly specific neurons in distinct subregions of the central amygdala. Pain *161*, 166–176. Doi: 10.1097/j.pain.0000000000001691.

Wilson TD, Valdivia S, Khan A, Ahn HS, Adke AP, Martinez Gonzalez S, Sugimura YK, Carrasquillo Y. (2019). Dual and opposing functions of the central amygdala in the modulation of pain. Cell Rep *29*, 332–346.e5. Doi: 10.1016/j.celrep.2019.09.011.

Ikeda R, Takahashi Y, Inoue K, Kato F. (2007). NMDA receptor-independent synaptic plasticity in the central amygdala in the rat model of neuropathic pain. Pain *127*, 161–172. Doi: 10.1016/j.pain.2006.09.003.

Penzo MA, Robert V, Tucciarone J, De Bundel D, Wang M, Van Aelst L, Darvas M, Parada LF, Palmiter RD, He M, Huang ZJ, Li B. (2015). The paraventricular thalamus controls a central amygdala fear circuit. Nature *519*, 455–459. Doi: 10.1038/nature13978.

5) The paw withdraw effect could be replicated by either reducing the CeA SST+ cell activity or by elevating PKCd+ cell activity, but this is not the case for the anxiety-like affective behaviors (Figure 4, Figure 4 S2). Therefore, the PGB effect on the increased paw withdraw threshold and emotion behaviors in the MP mouse model may be caused by distinct circuit mechanisms occurring simultaneously.To further elucidate their observation, the Author should test the acute PGB effect on the mechano-sensitivity and emotional behavior separately using a wild type mice or should at least acknowledge this possibility in the discussion.

Distinct circuit mechanisms could account for the increased paw withdraw threshold and improved emotion behaviors in the MP mouse model by PGB because PGB treatment is not specific to the inputs onto the CeA-SST neurons. Nevertheless, our Figure 2 —figure supplement 1 provides a parsimonious explanation for the effect of PGB on the increased paw withdraw threshold and emotion behaviors.

Regarding the PGB effect on mechano-sensitivity and emotional behavior, we plotted the changes of immobility time in the FST or number of buried marbles in the marble burying test against the change of mechano-sensitivity. Only the data obtained from the FST and marble burying test were used to address this question because these two behavioral assays could be tested before and after PGB application on the same animals. After PGB application, no positive correlation was observed between the change of the PW threshold and the change of the immobility time. A weak positive correlation between the change of PW threshold and the change in the number of buried marbles (Figure 2 —figure supplement 1), suggests that PGB alleviates the mechano-sensitivity and emotional behavior through distinct circuit mechanisms (see the Discussion page 9; line 16 – 33). For the EPM and social avoidance, the results can be influenced by repeated tests. Therefore, animals were only tested with either ACSF or PGB.

Reviewer #2 (Recommendations for the authors):My specific issues are outlined below:1) Why does the number of animals tested in the different behavioural paradigms (relative to Figures2 and 4) vary from test to test? Were these different cohorts for each behavioural test?

We indeed had different cohorts of mice for each behavioral test. This is somehow inevitable during the development of a project. For the same cohort, we randomly and equally assigned animals to the MP and Ctrl groups and then performed behavioral tests. Initially we focused on some common anxiety-like behavioral tests such as the EPM, L/D box, and marble burying tests. After obtaining positive results, we decided to add additional behavioral tests to strengthen the face validity of this pain model. Anxiety disorders are often associated with depression-like behavior and social avoidance. Therefore, we added the FST for depression and the three-chamber test for sociability. Addition of different cohorts for each behavioral test accounts for different numbers of animals in the different paradigms.

2) The size of the cohorts in which PGB was tested (and the relative controls) after induction of chronic muscle pain (Figure 2A-E) is suboptimal for the EPM and sociability tests, also in view of the known relatively high interindividual variability, and should, therefore, be increased.

We agree with the reviewer that the size of the cohorts is suboptimal for the EPM and sociability tests (Figure 2A–E). In revision, we increased the number of experiments in Figure 2B (Ctrl-ACSF, from n = 5 to 6; Ctrl-PGB, from n = 4 to 6; MP-ACSF, from n = 5 to 7; MP-PGB, from n = 6 to 7) and 2E (Ctrl-ACSF, from n = 5 to 6; Ctrl-PGB, from n = 5 to 7; MP-ACSF, from n = 4 to 6; MP-PGB, from n = 5 to 6). The following is the summary of the results.

Compared to the Ctrl mice, cannula infusion of PGB into the CeA of the MP mice increased the open-arm time in the EPM test (Ctrl: ACSF, 29.6 ± 3.8 %, n = 6; PGB, 20.0 ± 2.7 %, n = 5; Mann–Whitney test, U = 4, n.s., non-significant, *p* = 0.052. MP: ACSF, 15.1 ± 3.0 %, n = 7; PGB, 38.4 ± 5.1 %, n = 7; Mann–Whitney test, U = 2, ***p* = 0.002; Figure 2B) and increased the social-zone time in the three-chamber social test (Ctrl: ACSF, 44.6 ± 4.8 %, n = 6; PGB, 37.8 ± 3.8 %, n = 7; Mann–Whitney test, U = 10, n.s., non-significant, *p* = 0.138. MP: ACSF, 34.2 ± 2.9 %, n = 6; PGB, 48.6 ± 5.0 %, n = 6; Mann–Whitney test, U = 5, **p* = 0.041; Figure 2E). PGB at the dosage used in this study had no effect on the locomotion (Figure 2B; Ctrl: ACSF, 24.1 ± 2.9 m, n = 6; PGB, 22.9 ± 2.1 m, n = 5; Mann–Whitney test, U = 14, n.s., non-significant, p = 0.875. MP: ACSF, 21.4 ± 1.0 m, n = 7; PGB, 22.7 ± 2.3 m, n = 7; Mann–Whitney test, U = 18, n.s., non-significant, *p* = 0.446 Figure 2E; Ctrl: ACSF, 42.3 ± 2.3 m, n = 6; PGB, 35.2 ± 2.7 m, n = 7; Mann–Whitney test, U = 9, n.s., non-significant, *p* = 0.101. MP: ACSF, 39.5 ± 2.8 m, n = 6; PGB, 38.0 ± 3.6 m, n = 6; Mann–Whitney test, U = 15, n.s., non-significant, *p* = 0.675).

3) It should be made more clear in the text that in vivo ca^2+^ imaging experiments were performed in mice with chronic muscle pain. Why did the authors did not test the effect of PGB on SST+ neuron activity in control animals? This would have strengthened their findings. Moreover, they could have better exploited their study by testing the effect of PGB on SST+ neuron activity during a von Frey filament stimulation, as this test offers good behavioral bouts in which to analyse ca^2+^ transients.

Compared to the MP group, PGB had little effect on the mechanical sensitivity (Figure 1E, 1F) and affective behaviors (Figure 2B–E) in the Ctrl mice. Our finding suggests that PGB preferentially exerted its effect on the rewired pain/emotional circuits.

We do agree that the reviewer raised an important question. Compared to the rewired circuitry in the MP mice, the effect of PGB on neuronal activity in the Ctrl mice could be different. Considering that PGB had little effect on pain behavior in the Ctrl mice, we speculate that PGB could exert a marginal effect on SST neuron activity in the Ctrl mice. Nevertheless, we agree with the reviewer that testing the effect of PGB on SST+ neuron activity in the control mice would strengthen our findings and should be addressed in the future.

Regarding testing the effect of PGB on SST+ neuron activity during a von Frey filament stimulation, we agree that this experiment is interesting. Also, it may offer good behavioral bouts for detecting mechanically evoked ca^2+^ transients. In this study, we focused on maladaptive changes of pro-nociceptive neurons (that is, SST+ neurons) during the development chronic muscle pain. Therefore, we did not test the nociceptive response of the SST+ neurons in response to acute nociceptive stimulation such as acid injection.

Reviewer #3 (Recommendations for the authors):I have the following comments, which require more experiments to address in order to make the manuscript more convincing.1, Figure 3 shows recording results from CeA-SST and CeA-PKCd neurons in the MP mice. Since these neurons at baseline consist of different firing phenotypes (spontaneous, late-firing, and regular firing), it would be useful to know whether all firing types show the same changes in synaptic transmission and excitability in the MP model.Notably, in Wilson et al., 2019 paper, they found that CeA neurons require at least 50-80 pA current injection to induce the spikes, whereas in this manuscript, CeA neurons require a much lower current injection to fire. Could the author provide some explanation to this difference?

According to our previous study (Hou et al., 2016), we analyzed all CeA neurons shown in Figure 3 and classified them into early-spiking (ES) and late-spiking (LS) neurons based on their spike delay at rheobase. Using the same criteria, the CeA neurons with a spike delay greater than 1.5 s were classified as LS neurons, while the rest were classified as ES neurons (see Figure 3 —figure supplement 4). The ratio of these two subpopulations (ES vs. LS type) was not significantly altered in the MP mice.

For the changes in excitatory synaptic transmission (panel A and B in Figure 3 —figure supplement 4), compared to the Ctrl mice, sEPSC frequency was increased in SST-ES neurons of the MP mice (Ctrl, 1.3 ± 0.1 Hz, n = 4; MP, 3.5 ± 0.6 Hz, n = 9; Mann–Whitney test, U = 1, ***p* = 0.006). However, the sEPSC amplitude recorded from SST-ES neurons in the MP mice was not significantly different from that in Ctrl mice (Ctrl, 14.6 ± 3.0 pA, n = 4; MP, 16.9 ± 1.8 pA, n = 9; Mann–Whitney test, U = 14, *p* = 0.548). In contrast, the mean frequency (Ctrl, 2.8 ± 0.8 Hz, n = 5; MP, 4.9 ± 1.0 Hz, n = 11; Mann–Whitney test, U = 14, *p* = 0.138) and amplitude (Ctrl, 15.1 ± 2.6 pA, n = 5; MP, 16.9 ± 1.5 pA, n = 11; Mann–Whitney test, U = 23, *p* = 0.617) of SST-LS neurons in the MP mice were similar to those in the Ctrl mice.

On the other hand, sEPSC frequency in PKCδ-ES neurons of the MP mice was unchanged compared to the Ctrl mice (Ctrl, 4.5 ± 2.2 Hz, n = 3; MP, 1.7 ± 0.4 Hz, n = 7; Mann–Whitney test, U = 5, *p* = 0.267). Similarly, the sEPSC amplitude in PKCδ-ES neurons of the MP mice was not significantly different from that in the Ctrl mice (Ctrl, 17.1 ± 1.6 pA, n = 3; MP, 15.2 ± 1.3 pA, n = 7; Mann–Whitney test, U = 5, *p* = 0.267). In contrast, sEPSC frequency in PKCδ-LS neurons of the MP mice was significantly decreased compared to the Ctrl mice (Ctrl, 5.3 ± 1.0 Hz, n = 6; MP, 2.6 ± 0.3 Hz, n = 13; Mann–Whitney test, U = 13, **p* = 0.022). However, sEPSC amplitude in PKCδ-LS neurons of the MP mice was unchanged compared to the Ctrl mice (Ctrl, 16.3 ± 0.6 pA, n = 6; MP, 15.2 ± 1.3 pA, n = 13; Mann–Whitney test, U = 30, *p* = 0.47). Taken together, the upregulated excitatory transmission was specific to SST-ES neurons, whereas downregulated excitatory transmission was specific to PKCδ-LS neurons.

For the changes in cell excitability (panel C and D in Figure 3 —figure supplement 4), the SST-ES neurons in the MP mice generated more action potentials in response to depolarizing current injections compared to those in the Ctrl mice (Ctrl, n = 6; MP, n = 12; two-way ANOVA with Bonferroni’s multiple comparison test, F(1,78) = 10.84, ***p* = 0.0015). In agreement with increased excitability, the rheobase of SST-ES neurons in the MP mice was significantly lower than those in the Ctrl mice (Ctrl, 59.7 ± 9.5 pA, n = 6; MP, 34.2 ± 6.0 pA, n = 12; Mann–Whitney test, U = 11.5, **p* = 0.02). Intriguingly, the number of evoked spikes (Ctrl, n = 9; MP, n = 16; two-way ANOVA with Bonferroni’s multiple comparison test, F(1,114) = 1.618, *p* = 0.206) and rheobase of SST-LS neurons (Ctrl, 53.0 ± 8.4 pA, n = 9; MP, 42.1 ± 5.6 pA, n = 16; Mann–Whitney test, U = 52.5, *p* = 0.282) in the MP mice were similar to those in the Ctrl mice.

In great contrast to SST-ES neurons in the MP mice, the PKCδ-ES neurons in the MP mice generated less action potentials in response to depolarizing current injections compared to those in the Ctrl mice (Ctrl, n = 4; MP, n = 7; two-way ANOVA with Bonferroni’s multiple comparison test, F(1,45) = 16.4, ****p* = 0.0002). However, the rheobase of PKCδ-ES neurons in the MP mice was comparable to those in the Ctrl mice (Ctrl, 33.8 ± 3.1 pA, n = 4; MP, 63.0 ± 13.0 pA, n = 5; Mann–Whitney test, U = 3, *p* = 0.127). Similar to SST-LS neurons, the spike number (Ctrl, n = 7; MP, n = 13; two-way ANOVA with Bonferroni's multiple comparison test, F(1,88) = 2.84, *p* = 0.096) and rheobase (Ctrl, 30.8 ± 7.9 pA, n = 6; MP, 44.1 ± 4.0 pA, n = 12; Mann–Whitney test, U = 20.5, *p* = 0.157) of PKCδ-LS neurons in the MP mice were similar to those in the Ctrl mice. Taken together, the modification of cell excitability in the MP mice is specific to ES neurons in both CeA-SST and CeA-PKCδ neurons.

Regarding the difference in current threshold for spike generation, the rheobase was measured in the presence of synaptic GABAA (1 μM) and GABAB (1 μM) receptor blockers in our study. In contrast, spikes were evoked in the absence of synaptic blockers in the study by Wilson et al. (2019). Therefore, reduced inhibition in our recordings could account for the requirement of a much lower current injection for spike generation.

References:

Hou WH, Kuo N, Fang GW, Huang HS, Wu KP, Zimmer A, Cheng JK, Lien CC. (2016). Wiring specificity and synaptic diversity in the mouse lateral central amygdala. J Neurosci *36*, 4549–4563. Doi: 10.1523/JNEUROSCI.3309-15.2016.

Wilson TD, Valdivia S, Khan A, Ahn HS, Adke AP, Martinez Gonzalez S, Sugimura YK, Carrasquillo Y. (2019). Dual and opposing functions of the central amygdala in the modulation of pain. Cell Rep *29*, 332–346.e5. Doi: 10.1016/j.celrep.2019.09.011.

2, The number of animals in each test is quite different. For instance, in Figure 4E, n=10 for the mCherry group and n=52 for the hM4Di group; this is a big difference to compare 2 groups. Also, in Figure 4 supplement 2, in the CeA-PKCd experiment, n=4 for the mCherry group and n=13 for the hM3q group. These numbers are significantly fewer than those for CeA-SST hM4Di group (n=52). Please explain how the numbers of animals were assigned to each test? Did the author exclude some animals from some tests?

We designed two types of experiments to test the effect of silencing the CeA-SST neurons. Initially we tested the hM4Di effect on the MP mice (n = 43, Figure 4 —figure supplement 2). To this end, we expressed hM4Di on the SST neurons and then tested the effects of VEH (saline, i.p.) vs. CNO (i.p.) on the mechanical sensitivity in the MP mice. The result suggested that silencing the CeA-SST neurons by CNO reduces mechanical hypersensitivity. However, a potential caveat of this experiment is the off-target effect of CNO (Manvich et al., 2018). Therefore, we performed another group of experiments, in which we tested the effect of CNO on the mCherry- and mCherry-hM4Di- expressing MP mice (mCherry, n = 10; hM4Di, n = 9; Figure 4E).

In the originally submitted manuscript, we pooled the data obtained from the mCherry-hM4Di-expressing MP mice, which received the CNO treatment from the two experimental groups (n = 43 vs. 9). In revision, we separated the results of these two sets of experiment results. We only showed the CNO effect on the mCherry- and hM4Di-expressing MP mice (mCherry, n = 10; hM4Di, n = 9) in Figure 4E. Since the result is sufficient to support our conclusion, we moved the data, which compared the CNO vs. vehicle on the hM4Di-expressing MP mice, to the supplemental part (n = 43; Figure 4 —figure supplement 2).

References:

Manvich DF, Webster KA, Foster SL, Farrell MS, Ritchie JC, Porter JH, Weinshenker D. (2018). The DREADD agonist clozapine N-oxide (CNO) is reverse-metabolized to clozapine and produces clozapine-like interoceptive stimulus effects in rats and mice. Sci Rep *8*, 3840. Doi.org/10.1038/s41598-018-22116-z.

3, The authors did not show any post-hoc images of the virus injection/expression and co-localization with cell type specific markers for all the functional manipulation and imaging studies. These should be provided to substantiate the results.

In revision, we added images of virus expression (AAV5–hSyn-hM4Di-mCherry or AAV5–hSyn-hM3Dq-mCherry) colocalized with cell type specific markers (SST or PKCδ) in Figure 4B and Figure 4 —figure supplement 1A, 4B in the revised submission. The colocalization of specific markers with virus expression was summarized in Figure 4 —figure supplement 1B and 4B in the revised submission. Briefly, we injected an AAV5–hsyn-hM4Di-mCherry virus into the CeA of SST-Cre mice and performed SST immunohistochemistry (IHC) after virus expression (Figure 4B and Figure 4 —figure supplement 1). Our results revealed that 79.68 ± 1.87 % mCherry^+^ cells in the CeA are SST immunoreactive. Conversely, 63.63 ± 3.83 % SST-immunoreactive cells in the CeA are mCherry^+^ cells (SST+mCherry/SST, 63.63 ± 3.83 %; SST+mCherry/mCherry, 79.68 ± 1.87 %; n = 18 slice from two mice).

On the other hand, we injected an AAV5–hsyn-hM3Dq-mCherry virus into the CeA of PKCδ-Cre mice and performed PKCδ IHC after virus expression (Figure 4 —figure supplement 4B in the revised submission). Our results revealed that 62.73 ± 2.09 % mCherry^+^ cells in the CeA are PKCδ immunoreactive. Conversely, 34.11 ± 1.86 % PKCδ-immunoreactive cells in the CeA are the mCherry^+^ cells (PKCδ+mCherry/PKCδ, 34.11 ± 1.86 %; PKCδ+mCherry/mCherry, 62.73 ± 2.09 %; n = 21 slice from two mice).

4, In Figure 4 supplement 1B: Chemogenetic inhibition of CeA-SST neurons does not affect the paw withdrawal response in the naïve mice. In contrast, the authors themselves showed in their recent paper that optogenetic inhibition of CeA-SST neurons increased paw withdrawal response (Chen et al., 2022, Pain). Please reconcile these two different results in the same lab.

The reviewer pointed out a key issue regarding the use of in vivo optogenetic manipulations and interpretation of animal behavior. Indeed, we also performed optogenetic inhibition (5-min illumination) of CeA-SST neurons in the control mice in this study (data not shown in the submitted manuscript). The result showed that the mechanical sensitivity was reduced. It was different from our recent paper (Chen et al., 2022) that optogenetic inhibition (10-min illumination) of CeA-SST neurons increased mechanical sensitivity (that is, increased paw withdrawal response). It seems that different illumination protocols resulted in opposite behaviors. Several factors could confound interpretation of optogenetic experiments. First, illumination protocols and intracranial heating could activate or inactivate temperature-sensitive ion channels and thus bias animal behavior (Owen et al., 2019). Second, long-term optogenetic inhibition is known to cause accumulation of intracellular CI^-^, which could alter the E_GABA_ and reverse the GABAergic action (Raimondo et al., 2012). In our study, CeA-SST neurons receive reciprocal inhibition from neighboring CeA neurons. In our recent paper (Chen et al., 2022) and our unpublished data, yellow-light stimulation was applied to the naïve mice to activate the NpHR3.0 for 10 minutes. This could lead to intracellular Cl^-^ accumulation after termination of light pulse and cause an increase in excitability or rebound action potential generation in the CeA-SST neurons. Because of several experimental parameters of optogenetic manipulations can result in different behaviors, we decided to employ chemogenetic manipulations in this study.

References:

Owen SF, Liu MH, Kreitzer AC. (2019) Thermal constraints on in vivo optogenetic manipulations. Nat Neurosci *22*, 1061-1065. Doi: 10.1038/s41593-019-0422-3.

Raimondo JV, Kay L, Ellender TJ, Akerman CJ. (2012). Optogenetic silencing strategies differ in their effects on inhibitory synaptic transmission. Nat Neurosci *15*, 1102-1104. Doi:10.1038/nn.3143.

Chen WH, Lien CC, Chen CC. (2022). Neuronal basis for pain-like and anxiety-like behaviors in the central nucleus of the amygdala. Pain *163,* e463-e475. Doi:10.1097/j.pain.0000000000002389.

5, There is no data on a2d VGCC subunit expression in parabrachial or CeA in the chronic MP model in this manuscript. Given so much emphasis on PGB results, this information is crucial and is missing here.

Thanks for the reviewer’s comments. Cole et al., 2005 had described the anatomical distribution of α2δ VGCC subunit expression across the whole rat brain. It revealed that the α2δ VGCC subunit is expressed in the parabrachial nucleus and CeA. In the revised submission, we also provide published data by Allen Brain Institute (α2δ1: http://mouse.brain-map.org/experiment/show?id=72119649; α2δ2:http://mouse.brain-map.org/experiment/show?id=72119650), which show α2δ1 and α2δ2 VGCC subunit expression in the parabrachial in our supplement file (Figure 1 —figure supplement 2).

References:

Cole RL, Lechner SM, Williams ME, Prodanovich P, Bleicher L, Varney MA, Gu G. (2005). Differential distribution of voltage-gated calcium channel α-2 δ (alpha2delta) subunit mRNA-containing cells in the rat central nervous system and the dorsal root ganglia. J Comp Neurol *491*, 246-269. Doi:10.1002/cne.20693.

6, The authors performed the in vivo calcium imaging of CeA-SST neurons in response to PGB. It would be more informative to know how CeA-SST neurons or CeA-PKCd neurons respond to 1st and 2nd acid injection.

Indeed, it would be interesting to know how CeA-SST neurons or CeA-PKCδ neurons respond to 1^st^ and 2^nd^ acid injection. However, it is not the focus of this study. We speculate that several induction protocols including acidic saline injection or complete Freund’s adjuvant injection for chronic pain can differentially activate these two CeA neurons subtype. The latency of their activation in response to induction stimuli may be different. We plan to investigate how CeA-SST neurons or CeA-PKCδ neurons activate during induction in different types of chronic pain mouse models in the future.

[Editors’ note: further revisions were suggested prior to acceptance, as described below.]

Specifically:1) The text is often hard to read and should be carefully checked to improve readability. If needed, a professional writer could be helpful.

To improve the readability of the manuscript, we have submitted the manuscript to the Wiley Editing Services for professional editing.

2) All the reviewers acknowledged the additional excel database and your explanations regarding the data in Figure 3 – supplement 2 B-D. These data were submitted to support the electrophysiological results – that SST+ neuron excitability is selectively enhanced in MP mice. (See also comments of Reviewer 3 in their original review)However, some improvements are needed:a) We think that you pooled the data from "all slices" together at a given AP position relative to Bregma, to generate the plots and calculate SEM. However, we think you should re-calculate the mean at each AP position for each animal first, then average across animals and calculate SEM. In this way, the data would be averaged within an animal and the variance would be calculated between animals in order to assess the robustness of the results;

In the revised version of the manuscript, we conducted a new set of experiment (n = 5 animals in each group). As the reviewer suggested, we calculated the mean at each AP position for each animal first, then average across animals and calculate SEM (Figure 3 —figure supplement 2B, left). Similar to our previous results, the number of pERK cells was increased in the MP mice as compared to the Ctrl mice (Ctrl, 197.8 ± 16.5, n = 5; MP, 321.0 ± 30.6, n = 5; Mann–Whitney test, U = 1, **p* = 0.016; Figure 3 —figure supplement 2B, right). Notably, the total number of SST(+)pERK(+) in the CeA was increased in the MP mice (Ctrl, 8.2 ± 1.0, n = 5; MP, 22.0 ± 3.7, n = 5; Mann–Whitney test, U = 1, **p* = 0.016; Figure 3 —figure supplement 2C, right). In contrast, the total number of SST(-)pERK(+)cells in the CeA was slightly, but not significantly, increased in the MP mice (Ctrl, 189.6 ± 17.0, n = 5; MP, 298.8 ± 30.9, n = 5; Mann–Whitney test, U = 3, n.s., non-significant, *p* = 0.06; Figure 3 —figure supplement 2D, right).

b) Explain why the number of slices at a given AP position varies dramatically across animals (some animals have 1 slice, some have 5-9 slices at a given AP-position). Furthermore, how it could be that in some mice the AP position has 5-9 slices, whereas from the method one can calculate that a maximum of 3 slices can be obtained for a given AP position;

Thanks for pointing out this potential problem. The reason why some animals have 1 slice and some have 5-9 slices at a given AP position is because of the variation in the slice integrity and the slicing angle of each mouse. We agree with the reviewer that the number of slices should be consistent at a given AP position in an animal and should be less variable across animals. To this end, we conducted a new set of experiment and collected 2-5 slices at each AP position per mouse.

c)Tthe number of samples for each bregma level and animal should be similar and more control data should be collected and included since currently they show large variability;

In the revised version, we conducted a new set of experiment (n = 5 animals in each group) to have the same number of animals in each group.

d) Statistical analysis should be improved, e.g. a two-factor analysis of variance (with equal replications) would be a good way to analyze these data.

In the revised version, we conducted a new set of experiment. Both control and MP groups had equal numbers of animals. As the reviewer suggested, we used the two-way ANOVA with the Bonferroni's multiple comparison test to calculate the statistic power in the diary plot of panel B-D. Consistent to our previous results, the total number of pERK cells in the CeA along the AP axis was increased in the MP mice as compared to the Ctrl mice (Ctrl, n = 5; MP, n = 5; two-way ANOVA, F(1,64) = 29.66; *p* <0.0001 ; Figure 3 —figure supplement 2B, left). Similarly, the number of SST(+)pERK cells was increased in the MP mice as compared to the Ctrl mice (Ctrl, n = 5; MP, n = 5; two-way ANOVA, F(1,64) = 34.12; *p* <0.0001 ; Figure 3 —figure supplement 2C, left). Moreover, the numbers of SST(+)pERK(+) neurons at -1.46 and -1.58 AP position in the MP mice were significantly greater than those in the Ctrl mice (AP: -1.46: Ctrl, 1.33 ± 0.52, n = 5; MP, 3.74 ± 1.43, n = 5; two-way ANOVA with Bonferroni's multiple comparison test, F(1,64) = 34.12, **p* = 0.039; AP: -1.58: Ctrl, 0.68 ± 0.49, n = 5; MP, 3.06 ± 0.61, n = 5; two-way ANOVA with Bonferroni's multiple comparison test, F(1,64) = 34.12, **p* = 0.047; Figure 3 —figure supplement 2C, left). Likewise, the number of SST(-)pERK(+) cells in the CeA along the AP axis was increased in the MP mice as compared to the Ctrl mice (Ctrl, n = 5; MP, n = 5; two-way ANOVA, F(1,64) = 24.4; *p* <0.0001 ; Figure 3 —figure supplement 2D, left).

Finally, we took the comment of the reviewer 3 and compared the total number of cells in the CeA, instead of the averaged number across sections. Compared to the Ctrl mice, the total number of pERK cells was increased in the MP mice (Ctrl, 197.8 ± 16.5, n = 5; MP, 321.0 ± 30.6, n = 5; Mann–Whitney test, U = 1, **p* = 0.016; Figure 3 —figure supplement 2B). Similarly, the total number of SST(+)pERK(+) cells in the CeA was also increased in the MP mice (Ctrl, 8.2 ± 1.0, n = 5; MP, 22.0 ± 3.7, n = 5; Mann–Whitney test, U = 1, **p* = 0.016; Figure 3 —figure supplement 2C). In contrast, the total number of SST(-)pERK(+) cells was slightly, but not significantly, increased in the MP mice (Ctrl, 189.6 ± 17.0, n = 5; MP, 298.8 ± 30.9, n = 5; Mann–Whitney test, U = 3, n.s., non-significant, *p* = 0.06; Figure 3 —figure supplement 2D).

Reviewer #1 (Recommendations for the authors):I think that the Authors have done a good job in revising the manuscript according to my suggestions.

Thanks for the positive comment of the reviewer 1.

Reviewer #2 (Recommendations for the authors):The work by Lin and coworkers addresses the important and of broad interest process that underlies the progression of transient pain into persistent pain. This study used a number of sophisticated techniques to dissect the synaptic and cellular mechanisms that occur in the mouse central amygdala (CeA) during the development of chronic muscle pain. In particular, the study reveals shifts in opposite directions of the activity of somatostatin-expressing (SST+) and PKCδ+ neurons. Pharmacological and chemogenetic manipulation of these neurons prevented the chronification of pain and of the comorbid affective behaviors, although with important differences between these two neuronal classes.In their revised manuscript, the authors have made a number of significant changes by which they could adequately address my main concerns. In particular, the additional experimental data aimed at increasing the sample size of the behavioural experiments has appreciably improved the manuscript.While the text has improved, it still contains several syntax and grammar errors that need to be corrected.

Thanks to the positive comment of the reviewer 2. We have rechecked the manuscript and corrected the syntax and grammar errors. Furthermore, we have submitted the manuscript to the Wiley Editing Services for improvement of readability.

Reviewer #3 (Recommendations for the authors):I do have some questions for Figure 3—figure supplement 2In D, the number of SST(-)pERK(+) in the Ctrl (-1.82 mm) is above 40, while in B, the number of total pERK(+) in the Ctrl (-1.82mm) is about 30. This is weird. How could total number of pERK^+^ cells in B being less than that in D (a subset of neurons)?

In the previous version, we found that we mistakenly caused a slight shift of the control diary plot in the panel B. Together with the reviewer’s concern about the variability of slices at a given AP position and the different number of animals between the control and MP groups, we discarded the previous data and conducted a new set of experiment (see our response to comments in Essential revisions).

Authors concluded in Line 212-213, "increased pERK-positive cells are mostly CeA-SST cells." From the results shown in C and D, it's difficult to be convinced of this statement: the number of SST(+) pERK(+) cells increased from 1.1 to 3.7, while SST(-) pERK(+) cells increased from 18 to 28, even though statistics appeared to be not significant. I believe that instead of comparing the averaged number across sections, comparing the total number of cells in the CeA makes more sense.

For the new data, we took the reviewer’s comment and compared the total number of cells in the CeA. Compared to the Ctrl mice, the total number of pERK cells was increased in the MP mice (Ctrl, 197.8 ± 16.5, n = 5; MP, 321.0 ± 30.6, n = 5; Mann–Whitney test, U = 1, **p* = 0.016; Figure 3 —figure supplement 2B). Similarly, the total number of SST(+)pERK(+) cells in the CeA was also increased in the MP mice (Ctrl, 8.2 ± 1.0, n = 5; MP, 22.0 ± 3.7, n = 5; Mann–Whitney test, U = 1, **p* = 0.016; Figure 3 —figure supplement 2C). In contrast, the total number of SST(-)pERK(+) cells was slightly, but not significantly, increased in the MP mice (Ctrl, 189.6 ± 17.0, n = 5; MP, 298.8 ± 30.9, n = 5; Mann–Whitney test, U = 3, n.s., non-significant, *p* = 0.06; Figure 3 —figure supplement 2D).

The "gray" bars for "control" samples in panel B (total pERK^+^) and in panel D (SST-/pERK^+^) appear identical. Could there be an image duplication here?

In our old data, the "gray" bars for "control" samples in panel B (total pERK^+^) and in panel D (SST-/pERK^+^) appear similar, but not identical. Consistent with our old result, the diary plots and histograms in the new data indeed showed similar distribution (Figure 3 —figure supplement 2 – source data 1).